# Cancer-Associated Fibroblasts in Solid Tumors and Sarcomas: Heterogeneity, Function, and Therapeutic Implications

**DOI:** 10.3390/cells14171398

**Published:** 2025-09-07

**Authors:** Omar Badran, Idan Cohen, Gil Bar-Sela

**Affiliations:** 1Department of Oncology, Emek Medical Center, Afula 1834111, Israel; omar302702@gmail.com; 2Technion Integrated Cancer Center, Faculty of Medicine, Technion, Haifa 3525422, Israel; idan5161@gmail.com

**Keywords:** cancer-associated fibroblasts, tumor microenvironment, sarcoma, fibroblast heterogeneity, CAF-plasticity, mesenchymal tumors, stromal remodeling, tumor-stroma interaction, immune modulation, 3D tumor models

## Abstract

Cancer-associated fibroblasts (CAFs) are crucial regulators of the tumor microenvironment (TME), promoting cancer progression, immune suppression, and therapy resistance. Single-cell transcriptomics has identified at least five distinct CAF subtypes: myofibroblastic (myCAFs), inflammatory (iCAFs), antigen-presenting (apCAFs), metabolic (meCAFs), and vascular/developmental (vCAFs/dCAFs), each with unique localization, signaling, and functions. While CAFs are well studied in epithelial cancers, their roles in sarcomas are less understood despite the shared mesenchymal origin of tumor and stromal cells. This overlap blurs the line between malignant and non-malignant fibroblasts, raising fundamental questions about the identity of CAFs in mesenchymal tumors. In this narrative review, we explore the heterogeneity and plasticity of CAFs across solid tumors, focusing on their role in immune evasion, epithelial-to-mesenchymal transition (EMT), and resistance to chemotherapy, targeted therapy, and immunotherapy. We highlight emerging evidence on CAF-like cells in sarcomas and their contribution to tumor invasion, immune exclusion, and metastatic niche formation. We also assess new strategies to target or reprogram CAFs and suggest that CAF profiling may serve as a potential biomarker for patient stratification. Understanding CAF biology across various tumor types, including those with dense stroma and immunologically cold sarcomas, is crucial for developing more effective, personalized cancer treatments.

## 1. Introduction

In normal tissues, fibroblasts play a vital structural and regulatory role, maintaining the extracellular matrix (ECM) and supporting tissue homeostasis [1]. They are responsible for producing ECM proteins such as collagen and fibronectin and contribute to tissue repair after injury [2,3]. Under physiological conditions, fibroblasts stay quiescent but become temporarily activated during wound healing, adopting a contractile, myofibroblast-like phenotype that promotes matrix remodeling and secretes growth factors [4].

In the tumor microenvironment (TME), however, this activation becomes chronic and dysregulated [5]. Extended exposure to inflammatory signals, hypoxia, and tumor-derived cytokines reprograms normal fibroblasts into cancer-associated fibroblasts (CAFs), a diverse group of persistently active cells that support tumor growth [6,7]. Unlike temporary wound-healing fibroblasts, CAFs remain in an active state, continuously releasing pro-tumor factors, reshaping the extracellular matrix (ECM), promoting angiogenesis, and influencing immune responses to promote the growth, invasion, and immune evasion of cancer cells [8,9,10].

CAFs exhibit remarkable phenotypic plasticity and can originate from various sources, including resident fibroblasts, bone marrow-derived mesenchymal stem cells, pericytes, adipocytes, and even epithelial or endothelial cells through epithelial-to-mesenchymal transition (EMT) or endothelial-to-mesenchymal transition (EndMT) [11,12,13,14,15]. This diversity enhances their functional heterogeneity, as shown in recent single-cell RNA sequencing studies.

This review offers a comprehensive overview of CAF heterogeneity, function, and clinical significance across solid tumors, with a particular emphasis on their understudied roles in sarcomas. We investigate the origins of CAFs, their subtype-specific functions, and their roles in immune evasion, ECM remodeling, and therapeutic resistance. Because sarcomas originate from mesenchymal cells, the distinction between tumor cells and CAFs becomes unclear, raising important biological and clinical questions. We also review emerging strategies to target or reprogram CAFs and consider how understanding their context-dependent roles could enhance the effectiveness of cancer treatment.

## 2. Methods and Data Collection

We conducted a structured search of PubMed/MEDLINE, Embase, and Web of Science Core Collection from 1 January 2000 to 25 August 2025 (last search), focusing on English-language articles. Our research strategies combined controlled vocabulary (MeSH/Emtree) and free-text terms across three concept groups: (A) cancer-associated fibroblasts and subtypes (e.g., cancer-associated fibroblast/CAF, myCAF, iCAF, apCAF, meCAF/glyCAF, and vCAF/dCAF; FAP, α-SMA, and PDGFRα/β); (B) tumor types/entities (carcinomas and sarcomas, including PDAC, NSCLC, breast, gastric, osteosarcoma, and soft-tissue sarcoma); and (C) mechanistic/therapeutic pathways (e.g., TGF-β, CXCL12/CXCR4, Endo180/MRC2, LOX, NOX4, GLUT1, desmoplasia/ECM, immune-checkpoint blockade, and TLS). We supplemented database searches with the backward and forward citation chasing of key papers; Google Scholar was used exclusively for citation tracking. To contextualize interventional evidence, we queried ClinicalTrials.gov and the EU Clinical Trials Register, considering only trials with peer-reviewed results.

Eligibility criteria and scope: We included human clinical studies (randomized trials, prospective/retrospective cohorts, and case–control studies), translational analyses of human tumors, preclinical in vivo/in vitro studies, and high-quality reviews, all published in English between 2000 and 2025, focused on solid tumors in carcinomas and sarcomas (adult and pediatric). Eligible exposures were CAF programs/subtypes (myCAF, iCAF, apCAF, meCAF/glyCAF, and vCAF/dCAF), stromal checkpoints (e.g., Endo180, TNC), and CAF-targeted axes (e.g., TGF-β, CXCL12/CXCR4, NOX4, LOX, and GLUT1). Outcomes included mechanistic/translational readouts (e.g., intratumoral CD8 access, chemokine gradients, ECM stiffness/LOX activity, and perfusion/hypoxia) and clinical endpoints (response, PFS/OS, and ICB benefit/resistance). Modalities encompassed scRNA-seq, spatial multi-omics, multiplex IHC/IF, bulk RNA/proteomics, and functional perturbations. For sarcoma-focused studies, we noted when lineage/copy-number-aware methods were used to distinguish stromal fibroblasts from CAF-like tumor cells.

### 2.1. Study Selection, Data Extraction, and Appraisal

Records were de-duplicated and screened in two stages (title/abstract, then full text) by two independent reviewers (O.B., I.C.); disagreements were resolved by consensus or third-author adjudication (G.B.-S.). From each study we recorded tumor type, model/system, CAF program or subtype markers, spatial context, stromal readouts (ECM stiffness/LOX, perfusion/hypoxia), immune metrics (e.g., CD8 density/exclusion), therapeutic axis, and the qualitative direction of effect. For human interventional and observational studies, we noted basic indicators of rigor when reported (e.g., randomization/blinding, sample-size statements, and clarity of endpoints). For animal studies, we similarly noted randomization/blinding and endpoint definition. In mechanistic papers, we flagged recurring interpretation pitfalls (e.g., lineage misclassification of malignant mesenchymal cells as “CAFs,” loss of ECM-anchored CAFs during dissociation, and spatial claims without spatial validation).

### 2.2. Synthesis and Bias Mitigation

We conducted a qualitative narrative synthesis only; no quantitative pooling or meta-analysis was performed. Given the heterogeneity in designs, populations, and endpoints, findings are integrated by biological theme and context, with neutral/negative results included where identified. To reduce citation bias, we used broad, multi-database searches, bidirectional citation chasing, and limited self-citation; preprints without peer review were excluded.

### 2.3. Limitations

This is a structured narrative review (not PRISMA-registered). Some studies may have been missed, and no formal quantitative synthesis was attempted. Conclusions reflect the qualitative integration of heterogeneous mechanistic and translational evidence.

## 3. Subtypes of Cancer-Associated Fibroblasts (CAFs)

The traditional view of CAFs as a uniform group has changed significantly over the past decade. Advances in single-cell RNA sequencing and spatial transcriptomics have revealed that CAFs exhibit considerable diversity in both appearance and function. Researchers have identified at least five main CAF subtypes, each playing different roles in tumor biology based on their location, surface markers, secretory profiles, and interactions with other parts of the TME [16,17].

### 3.1. Myofibroblastic CAFs (myCAFs)

Myofibroblast cancer-associated fibroblasts (myCAFs) are a specific subset of CAFs characterized by their high expression of α-smooth muscle actin (α-SMA) and their spatial localization near tumor cells [18,19,20]. These cells exhibit a contractile phenotype like that of wound-healing myofibroblasts, playing a crucial role in remodeling the ECM [4,21]. By secreting collagen, fibronectin, and lysyl oxidase, myCAFs contribute to the desmoplastic reaction, which increases tissue stiffness and forms physical barriers that support tumor invasion and hinder the delivery of drugs [22,23]. Importantly, this ECM remodeling is not only structural; it also influences mechanotransduction pathways in cancer cells, such as the Yes-Associated Protein and Transcriptional Co-Activator with PDZ-binding motif (YAP/TAZ) signaling, which promotes proliferation, migration, and therapy resistance [24]. myCAFs also frequently express high levels of transgelin (TAGLN), an actin-binding protein (SM22) encoded on chromosome 11q23.2 that regulates cytoskeletal organization, contractility, and extracellular matrix remodeling [25]. TAGLN expression in myCAFs supports their activated, contractile phenotype and contributes to tumor progression. In colorectal cancer (CRC), TAGLN^+^ fibroblasts correlated with high EMT scores, advanced stage, poor prognosis, and resistance to immunotherapy [25]. At the same time, TAGLN knockdown reduced tumor cell proliferation, migration, and EMT phenotypes in vitro and suppressed tumor growth in vivo [25]. In NSCLC, myCAFs are α-SMA–high stromal cells that remodel the ECM, creating a stiff, pro-tumorigenic environment that drives tumor growth, invasion, and therapy resistance. They promote chemoresistance by blocking drug penetration and secreting resistance factors, thereby reducing EGFR-TKI efficacy through bypass signaling, and impairing immunotherapy responses through immune exclusion and TGF-β–mediated suppression. MyCAFs also recruit immunosuppressive cells, form metastasis-supportive niches, and display organ-specific roles in spread. Targeting their activation pathways, ECM production, or specific markers offers therapeutic potential to improve treatment outcomes in. Elevated stromal TAGLN was associated with lymphatic metastasis and enhanced fibroblast activation and motility, which was mediated in part by NF-κB pathway activation and IL-6 secretion, thereby fostering a pro-inflammatory, tumor-promoting microenvironment. Collectively, TAGLN in myCAFs plays a pivotal role in metastasis, immune evasion, and therapy resistance, identifying it as both a prognostic biomarker and a potential therapeutic target [26].

Despite their pro-tumor genic roles, myCAFs have also been associated with tumor suppression in certain situations [27]. In genetically engineered mouse models of pancreatic ductal adenocarcinoma (PDAC), removing α-SMA+ myoCAFs resulted in decreased survival, increased immunosuppressive infiltrates such as regulatory T cells, and tumors that were less differentiated and more aggressive [28,29]. This paradox highlights the adaptability of myCAFs and underscores their ability to both promote and inhibit tumor growth depending on the molecular and spatial context. In urothelial carcinoma (UC), the myCAFs can promote stromal remodeling, cancer stemness, and immune exclusion, and are associated with higher stage, poorer prognosis, and resistance to immunotherapy. In gastric cancer (GC), myCAFs are a key stromal component, marked by high α-SMA, FAP, and Collagen I expression. Single-cell RNA sequencing has revealed a bidirectional interaction between myCAFs and a specific GC cell program (MP7), creating a vicious cycle that fosters an immunosuppressive tumor microenvironment and promotes metastasis.

Recent single-cell transcriptomic studies have revealed that myCAFs are not a single, uniform group, but rather include subclusters with distinct gene expression patterns. For example, in breast cancer, different subsets of ECM-myCAFs and Transforming Growth Factor Beta (TGFβ)-myCAFs have been identified; both help tumors evade the immune system and resist immunotherapy by decreasing CD8-positive T cells (CD8+ T) cell infiltration and increasing TGF-β signaling [30,31]. Additionally, myCAFs may facilitate the development of immune-excluded tumor types, which is a common feature of resistance to immune checkpoint blockade therapies [32,33].

### 3.2. Inflammatory CAFs (iCAFs)

Inflammatory cancer-associated fibroblasts (iCAFs) are a phenotypically and functionally distinct subset of cancer-associated fibroblasts [34]. Inflammatory cancer-associated fibroblasts (iCAFs) are a pro-inflammatory subset of tumor-associated fibroblasts characterized by high secretion of cytokines and chemokines, most notably CXCL1. Signals from the tumor microenvironment, such as reduced TGF-β signaling, activation of NF-κB and STAT3 pathways, or stimulation by oncostatin M from tumor-associated macrophages and neutrophils, strongly upregulate CXCL1 expression in iCAFs. CXCL1 released by iCAFs promotes multiple pro-tumorigenic processes: it recruits neutrophils and myeloid-derived suppressor cells (MDSCs) that dampen anti-tumor immunity, stimulates both direct and indirect angiogenesis, and induces epithelial-to-mesenchymal transition (EMT), migration, and cancer stem cell (CSC) expansion in tumor cells (Figure 1). In many malignancies—particularly in carcinomas such as triple-negative breast cancer (TNBC), cervical cancer, platinum-resistant ovarian cancer, and advanced prostate cancer—iCAFs are a primary stromal source of CXCL1, thereby contributing to aggressive tumor biology, metastasis, and therapy resistance. This makes the iCAF–CXCL1 axis an attractive therapeutic target aimed at modulating the tumor microenvironment. Unlike contractile myCAFs, iCAFs are usually located farther from tumor cells and are transcriptionally activated by Interleukin-1 (IL-1) through the Janus Kinase/Signal Transducer and Activator of Transcription (JAK/STAT) signaling pathway [35,36]. Conversely, TGF-β signaling suppresses this phenotype and promotes the differentiation of myCAFs [30,37]. This reciprocal signaling dependence highlights the dynamic plasticity of CAF states, allowing transitions between iCAF and myCAF identities in response to microenvironmental cues [38].

Functionally, iCAFs play a key role in creating an immunosuppressive TME. The cytokines secreted by iCAFs can attract myeloid-derived suppressor cells (MDSCs) and direct tumor-associated macrophages (TAMs) toward an M2-like phenotype, which then suppresses cytotoxic T cell activity and promotes immune evasion [39,40]. IL-6, in particular, is a powerful driver of Signal Transducer and Activator of Transcription 3 (STAT3) activation in both immune and tumor cells, supporting chronic inflammation, resistance to cell death, and the preservation of cancer stem cell traits [41,42]. Additionally, CXCL12 secretion by iCAFs creates a physical and chemotactic barrier that prevents CD8+ T cells from reaching the tumor core, aiding resistance to immune checkpoint inhibitors [43,44].

In GC, iCAFs are a pro-tumorigenic fibroblast subtype. They have been shown to play a carcinogenic role in diffuse GC and may be involved in the de novo carcinogenesis. In skin cancers, particularly infiltrative BCC and aggressive melanomas, iCAFs are markedly increased, correlating with higher malignancy and aggressiveness. These iCAFs drive an inflammatory microenvironment that supports tumor growth and spread by producing most of the cytokines, chemokines, and immunomodulatory factors that recruit and activate neutrophils, T cells, and NK cells. Notably, normal dermal fibroblasts can be reprogrammed into an iCAF-like phenotype when exposed to skin cancer cell secretions, mirroring the chemokine/cytokine profile seen in tumor-associated iCAFs. Recent single-cell RNA sequencing (ssRNA-Seq) studies in pancreatic and breast cancers have further refined the iCAF phenotype, distinguishing subclusters based on differences in cytokine expression profiles and spatial distribution [45]. In some models, iCAFs have also been linked to triggering EMT through paracrine TGF-β and IL-6 signaling, encouraging invasion and metastatic potential [46].

### 3.3. Antigen-Presenting CAFs (apCAFs)

Antigen-presenting cancer-associated fibroblasts (apCAFs) are a distinct subset of fibroblasts within carcinomas that express MHC class II molecules together with CD74. This invariant chain stabilizes MHC-II and facilitates antigen loading. This phenotype enables apCAFs to present processed antigens to CD4+ T cells within the tumor microenvironment, thereby establishing direct antigen-specific interactions [47,48]. However, despite their antigen-presenting capacity, apCAFs typically exhibit low or absent expression of co-stimulatory molecules such as CD80, CD86, and CD40, which are essential for full T cell activation [47,48]. As a result, MHC-II+/CD74+ apCAFs may drive divergent immune outcomes: in some contexts, they induce the differentiation of naïve CD4+ T cells into immunosuppressive regulatory T cells (Tregs), thereby enhancing immune evasion; in others, they can stimulate effector CD4+ T cell responses that contribute to anti-tumor immunity. This dual functionality highlights the complex and context-dependent role of MHC-II+/CD74+ apCAFs in shaping tumor immunity, positioning them as both potential targets and modulators in immunotherapeutic strategies [47,48]. Initially identified in PDAC through ssRNA-Seq, apCAFs were shown to interact with Cluster of Differentiation 4-positive T lymphocytes (CD4^+^ T cells) via MHC–II-dependent mechanisms, potentially presenting antigens and influencing T cell responses without full activation due to the absence of co-stimulation [48,49]. In NSCLC patients receiving neoadjuvant chemoimmunotherapy, apCAFs are linked to unfavorable outcomes. Their expansion is driven by interferon-γ through the JAK1/2-STAT1 pathway, leading to increased expression of PD-L2 and activation of the PD-L2-RGMB axis. This process promotes the formation of immunosuppressive FOXP1+ regulatory T cells, which weaken the anti-tumor immune response. Blocking the PD-L2-RGMB axis has been shown to reduce apCAF-mediated Treg accumulation and enhance the efficacy of immunotherapy. In colCRC, apCAFs can stimulate CD4+ T cell activation, evidenced by increased CD69 and CD25 expression, when presenting mutated SLP to mismatched fibroblast cell lines. However, this same interaction reduces CD8+ T cell cytotoxicity and activation, indicating a dual role in both promoting helper T cell responses and dampening cytotoxic T cell activity.

Recent studies have expanded the significance of apCAFs beyond PDAC, identifying them in NSCLC and GC. apCAFs with immunostimulatory features have been reported in non-small cell carcinoma (NSCLC) adjacent to CD4^+^ T cells, suggesting a context-dependent function. In GC, apCAFs were found to be enriched in tumor regions near tertiary lymphoid structures (TLS), indicating a role in supporting local immune cell activation and organization [50]. Spatial transcriptomics and immunohistochemistry analyses have confirmed their colocalization with T- and B-cell clusters, and their abundance is associated with improved responses to immune checkpoint blockades [50]. Mechanistically, apCAFs have been shown to enhance the activation and killing ability of CD4^+^ T cells, as well as promote M1-like macrophage polarization, thereby creating a positive feedback loop that boosts anti-tumor immunity [50,51]. However, the duality of the apCAF function remains an area of investigation. In specific settings, the absence of co-stimulatory molecules may lead to T cell anergy or the differentiation of regulatory T cells (Tregs), potentially suppressing immunity [52,53].

### 3.4. Metabolic CAFs (meCAFs)

Metabolically active cancer-associated fibroblasts (meCAFs) form a specific subset of CAFs identified by their key role in enhancing tumor metabolism through reprogramming [54]. Unlike myCAFs or iCAFs, meCAFs primarily exhibit increased levels of glycolytic enzymes, glucose transporters (GLUT1), lipid metabolism enzymes, and amino acid transporters, indicating their active involvement in modifying the tumor’s metabolic environment [5].

meCAFs participate in a “reverse-Warburg” coupling in which stromal glycolysis fuels tumor OXPHOS (via lactate and other metabolites). The Warburg effect describes how cancer cells prefer aerobic glycolysis over oxidative phosphorylation, leading to increased glucose intake and lactate production even when oxygen is available [55]. This metabolic change supports rapid tumor growth and creates an acidic microenvironment that influences CAF activation and function [55]. These metabolites are absorbed by cancer cells and used in the tricarboxylic acid (TCA) cycle to generate energy through oxidative phosphorylation (OXPHOS), thereby encouraging tumor progression [56]. Consequently, CAFs in breast and lung tumors display higher levels of glycolytic enzymes, including lactate dehydrogenase (LDH), pyruvate kinase M2 (PKM2), and the lactate transporter Monocarboxylate Transporter 4 (MCT4) [57,58]. Lactate acts as an energy shuttle between stromal and cancer cells, similar to its role in the brain and heart [59,60,61]. This reverse Warburg-like metabolic interaction promotes tumor growth, invasiveness, and resistance to therapy [62]. Also, elevated expression levels of genes involved in the lactate shuttle system, particularly high expression of the monocarboxylate transporter 4 (MCT4), are associated with poor prognosis in prostate cancer and pancreatic cancer. These meCAFs have very active metabolism, particularly in glycolysis, which enables them to sustain cancer cell growth in nutrient-deficient or hypoxic tumor regions by providing alternative energy sources and buffering the acidic microenvironment. They are linked to increased metastasis and poorer prognosis [63]. Notably, in a study examining them in the context of PDAC, tumors with abundant meCAFs showed markedly better responses to immunotherapy and greater infiltration of CD8+ T cells [64]. 

Mechanistically, the development of meCAF phenotypes is driven by paracrine signals, such as IL-6, TGF-β, and tumor-derived exosomal miRNAs (e.g., miR-105), which activate the Myelocytomatosis oncogene (MYC) and Ataxia Telangiectasia Mutated (ATM) signaling pathways in fibroblasts, thereby causing metabolic shifts [65]. However, in breast cancer, high meCAF activity has been linked to resistance to chemotherapy and immune checkpoint blockade, due to both metabolic support and the promotion of an immunosuppressive microenvironment through lactic acid-induced T cell dysfunction and macrophage polarization [66,67].

### 3.5. Vascular and Developmental CAFs (vCAFs and dCAFs)

Among the recently identified subsets of cancer-associated fibroblasts, vascular cancer-associated fibroblasts (vCAFs) and developmental cancer-associated fibroblasts (dCAFs) are specialized groups with distinct transcriptional profiles and locations within the tumor microenvironment. vCAFs originate from perivascular stromal cells and are found near tumor blood vessels [68]. These cells exhibit high levels of pro-angiogenic mediators, such as Vascular Endothelial Growth Factor A (VEGFA), Angiopoietin-like 4 (ANGPTL4), and IL-6, along with matrix-remodeling enzymes like matrix metalloproteinase 9 (MMP9), which aid in endothelial cell movement and new vessel formation [69]. Functionally, vCAFs remodel and often aberrantly stabilize tumor vessels; this can increase local perfusion while sustaining leaky/dysfunctional vasculature that supports growth and therapy resistance [70].

Vascular and developmental CAFs (vCAFs and dCAFs) are two distinct fibroblast subtypes within the tumor microenvironment, first resolved by single-cell RNA sequencing in mammary tumor models. They are defined by where they sit in the tumor, how they likely arise, and what they do functionally. Together, they illustrate that CAFs are not a single entity but a spectrum of stromal states with specialized roles in cancer progression. vCAFs occupy the perivascular niche, hugging tumor blood vessels and likely deriving from perivascular stromal cells and pericytes. Functionally, they are pro-angiogenic: they secrete factors that promote new vessel growth and remodeling, and a characteristic vCAF transcriptional signature has been associated with poor prognosis and metastasis in mammary cancer. Marker-wise, vCAFs show high PDGFRβ alongside perivascular/endothelial-adjacent programs, reflecting their vascular coupling. dCAFs, by contrast, localize closer to tumor cells and carry a developmental or progenitor-like expression program. They have been linked to malignant cells that underwent epithelial-to-mesenchymal transition (EMT), and they express genes associated with developmental states and stemness (e.g., Sox9; sometimes Scgr1/Sox10 depending on dataset). Functionally, dCAFs are implicated in supporting invasive growth and broader tumor progression through matrix remodeling and developmental signaling axes. In cholangiocarcinoma (ICC), vCAFs are associated with proliferation and microvasculature gene signatures. IHC staining in intrahepatic CCA samples revealed CD146+ vCAFs in both the tumor core and microvascular regions, indicating a potential key role in interactions with malignant cells.

Importantly, vCAFs and dCAFs coexist with other CAF states inside the same tumor, and CAFs can arise from multiple sources, resident fibroblasts, mesenchymal precursors, or even epithelial/endothelial cells after transition. This heterogeneity is dynamic: niches, mechanical stress, hypoxia, and therapy can shift the balance among CAF states over time. Understanding these differences is crucial for therapy design, as anti-angiogenic or vessel-normalizing strategies may be most effective against vCAF-driven programs. In contrast, dCAF-rich contexts may require agents that modulate developmental/EMT pathways, as well as extracellular matrix remodeling.

### 3.6. Tumor-Promoting Versus Tumor-Suppressive Functions of CAFs

CAFs show a functional split within the TME, acting either as promoters or suppressors of tumor growth [71]. This dual role results from the wide variety and adaptability of these cells, which are affected by spatial location, tumor type, and ongoing cellular interactions. Most studies have historically focused on the tumor-promoting activities of CAFs, which include facilitating cancer cell invasion, immune evasion, angiogenesis, and therapeutic resistance.

Tumor-promoting CAFs (pCAFs) actively support carcinogenesis through various mechanisms. They remodel the ECM by producing structural proteins such as collagen and fibronectin, along with Matrix Metalloproteinases (MMPs), which degrade ECM components, increase tissue stiffness, and create invasion pathways for tumor cells [72]. The production of MMPs enables CAFs to promote further invasion of cancer cells, making these enzymes viable therapeutic targets [72]. In NSCLC, ECM remodeling and the secretion of growth factors by CAFs contribute to increased tissue stiffness, which enhances the adhesion of metastatic cancer cells to the tumor endothelium, thereby exacerbating metastatic progression [72]. Additionally, pCAFs secrete a variety of cytokines and chemokines, including Interleukin-6 (IL-6), CXCL12, and transforming growth factor-beta (TGF-β), which inhibit cytotoxic T-lymphocyte activity and attract immunosuppressive cells such as regulatory T cells (Tregs), TAMs, and MDSCs [73,74]. These CAFs also promote angiogenesis by releasing Vascular Endothelial Growth Factors (VEGFs) and supplying metabolic substrates like lactate and glutamine to cancer cells, thereby supporting their metabolic reprogramming and growth [75]. Moreover, CAFs contribute to resistance against chemotherapy and immunotherapy by strengthening physical barriers within the tumor stroma and maintaining an immunosuppressive environment [76]. For instance, in pancreatic ductal adenocarcinoma, PDAC, CAF-secreted CXCL12 (SDF-1), has been reported to increase SATB1 expression in tumor cells and to contribute to gemcitabine resistance [77].

In contrast, a subset of CAFs has tumor-suppressive properties. These tumor-restraining CAFs (rCAFs) can act as physical barriers, thereby limiting tumor spread and preserving tissue structure [9,78,79]. Recent studies indicate that specific subsets of CAFs, especially those producing Cluster of Differentiation 9 (CD9)-positive exosomes, can inhibit melanoma cell growth and are associated with improved long-term survival in patients [80]. In NSCLC, Cluster of Differentiation 200 (CD200)-expressing CAFs have been found to increase tumor cell sensitivity to targeted therapies, such as Epidermal Growth Factor Receptor (EGFR) inhibitors, and this effect disappears when CD200 is absent [81]. Other evidence suggests that factors secreted by CAFs, including Insulin-like Growth Factor (IGF) and IGF-binding proteins, may enhance the response to drugs in NSCLC [82,83]. In PDAC, the removal of specific myCAF populations has been linked to increased immunosuppression and accelerated tumor growth, suggesting that these cells may serve a protective role [84]. Additionally, antigen-presenting CAFs (apCAFs) may assist in activating immune responses against tumors [84]. These findings emphasize the complexity of CAF biology and underscore the need for further research into their context-dependent, anti-tumor functions. Cancer-associated fibroblasts (CAFs) play dynamic and evolving roles in the progression of tumors. They adapt their phenotype and functions as the tumor microenvironment (TME) changes at various stages of cancer development.

The “good vs. bad CAF” story makes sense once context, the fibroblast state, spatial niche, and tumor stage are considered. Myofibroblastic CAFs (myCAFs) are α-SMA-high, TGF-β-responsive stromal cells that reside close to tumor nests and are major producers of the extracellular matrix (e.g., collagen), generating a dense, stiff stroma that impedes drug delivery and CD8+ T cell trafficking; accordingly, tumors enriched for myCAFs often show an abundant ECM and worse prognosis. Yet the biology is dual: in preclinical models, non-selective depletion of myCAFs—or the removal of the type I collagen they deposit—can paradoxically hasten tumor progression and heighten immunosuppression (including MDSC recruitment). Together, these data indicate that myCAFs exert context-dependent and both tumor-promoting and tumor-restraining effects, arguing for selective reprogramming/normalization of myCAF states rather than indiscriminate stromal ablation.

Inflammatory CAFs (iCAFs) are IL-1-induced, α-SMA-low, IL-1R-high fibroblasts that typically localize farther from tumor nests. They activate the JAK/STAT and NF-κB pathways and secrete a broad cytokine/chemokine milieu (e.g., IL-6, IL-1, CXCL1, CXCL12/SDF-1, HGF, and CCL17) that recruits and polarizes MDSCs and M2-like macrophages while also expanding Tregs. This process suppresses cytotoxic T cell activity and promotes tumor growth. iCAFs and myCAFs are interconvertible states—IL-1 drives iCAF features, whereas TGF-β pushes toward myCAF—underscoring CAF plasticity. Their positioning beyond the dense, myCAF-built ECM may aid the diffusion of immunosuppressive signals, reinforcing immune exclusion without requiring direct proximity to cancer cells.

Antigen-presenting CAFs (apCAFs) are MHC-II^+^/CD74^+^ fibroblasts that typically lack co-stimulatory molecules (CD80/CD86/CD40), so in many settings they skew naïve CD4^+^ T cells toward regulatory T cells rather than complete activation. In PDAC, apCAFs appear to arise from mesothelial cells that, under the influence of IL-1 and TGF-β, downregulate mesothelial features and acquire fibroblastic traits; their spatial patterning is less stereotyped than that of iCAFs/myCAFs in some tumors. Notably, context matters: in human NSCLC, apCAFs have been observed adjacent to—and capable of activating—CD4^+^ T cells, suggesting that even within the “apCAF” label, function can range from tolerogenic (Treg-inducing) to immunostimulatory, depending on tumor type and microenvironment.

meCAFs are fibroblasts rewired toward high glycolysis (e.g., GLUT1, LDH, PKM2, and MCT4), exporting lactate and other metabolites that acidify the TME and “fuel” neighboring cancer cells (reverse-Warburg coupling). They help sustain growth in hypoxic, nutrient-poor regions and are induced by paracrine cues (IL-6, TGF-β) and tumor exosomal miRNAs. Functionally, meCAFs are linked to immune dysfunction (lactate-driven T cell impairment, macrophage polarization) and resistance to chemotherapy and ICB. Therapeutically, inhibiting glycolytic flux or transport can reduce chemokine barriers, restore T cell access, and enhance the efficacy of systemic therapy. In PDAC, the link between fatty acid synthase (FASN) expression in the tumor stroma is primarily associated with meCAFs (metabolically activated CAFs) rather than the entire CAF population. These cells are typically located in less densely fibrotic regions of the tumor, where reduced mechanical stress allows for enhanced metabolic activity. Through FASN-driven de novo lipogenesis, meCAFs supply pancreatic cancer cells with fatty acids that support membrane synthesis, energy production, and redox balance, thereby promoting tumor proliferation, survival, and resistance to therapy such as gemcitabine. In contrast, myCAFs (myofibroblastic CAFs) are more focused on extracellular matrix remodeling and display lower lipid metabolic activity, while iCAFs (inflammatory CAFs) mainly secrete cytokines and chemokines. This highlights meCAFs as a metabolically active stromal niche that fuels PDAC progression via FASN-dependent lipid transfer, positioning the meCAF–FASN axis as a promising target for therapeutic intervention (Figure 1).

vCAFs originate from perivascular stromal cells and localize near tumor vessels, secreting pro-angiogenic factors (e.g., VEGFA, ANGPTL4) and matrix remodelers (e.g., MMP9), which stabilize abnormal vasculature, improve the nutrient supply to tumors, and contribute to therapy resistance. dCAFs represent CAFs with developmental/embryonic gene programs; although less well defined, they are associated with tissue remodeling, growth cues, and invasive phenotypes. Together, vCAFs and dCAFs highlight vascular and developmental axes of stromal support that may be targetable alongside anti-angiogenic or matrix-normalizing strategies.

Because CAF subtypes are heterogeneous, plastic, and context-dependent, effective therapy should favor selective, biomarker-guided reprogramming or axis blockade (matrix, metabolic, and immune) rather than indiscriminate stromal depletion.

## 4. Crosstalk with Cancer Stem Cells (CSCs)

CAFs also play a crucial role in maintaining cancer stem cells (CSCs), a subpopulation of tumor cells characterized by their ability to self-renew and exhibit pluripotency [1]. They are responsible for producing ECM proteins such as collagen and fibronectin and contribute to tissue repair after injury [2,3]. Under physiological conditions, fibroblasts remain quiescent but become temporarily activated during wound healing, adopting a contractile, myofibroblast-like phenotype that is associated with a high metastatic potential [85,86]. The interaction between CAF and cancer stem cells (CSC) is facilitated through a complex network of secreted factors, signaling molecules, and extracellular vesicles that work together to enhance the stem-like qualities of cancer cells [87].

IL-6 secreted by CAFs activates the Janus Kinases/Signal Transducer and Activator of Transcription 3 (JAK/STAT3) signaling pathway in tumor cells, thereby enhancing stemness, EMT, and resistance to apoptosis [88,89]. Additionally, CAFs secrete Wingless/Integrated signaling protein (WNT) ligands and TGF-β, both of which are essential for maintaining the CSC phenotype [90]. WNT signaling promotes β-catenin activation, while TGF-β facilitates EMT and niche adaptation, fostering metastatic ability [90,91]. Beyond soluble factors, CAFs also communicate with CSCs through the release of exosomes [92,93]. These extracellular vesicles can carry various bioactive molecules, including microRNAs (miRNAs), metabolites, and proteins that promote stemness and therapeutic resistance [92,93]. For example, CAF-derived exosomes enriched with miR-21 or miR-146a have been shown to enhance CSC traits and resistance to chemotherapeutic drugs [94,95]

Additionally, the close physical proximity between CAFs and CSCs in the tumor microenvironment supports the notion that CAFs play a crucial role in the CSC niche [96]. Through direct cell-to-cell contact and remodeling of the ECM, CAFs contribute to creating a microenvironment that protects CSCs from immune surveillance and the stress caused by treatments, thereby promoting tumor recurrence and metastasis [97].

## 5. CAFs in Sarcomas

Sarcomas are mesenchymal malignancies that develop from bone, muscle, fat, cartilage, or connective tissue. They are biologically and clinically diverse, and they differ significantly from carcinomas, which originate from epithelial cells. This mesenchymal origin influences the tumor microenvironment (TME), including the extracellular matrix (ECM), vascularity, and immune context, and affects how stromal programs interact with tumor cells [98,99] (Figure 1).

Fibroblasts are resident stromal cells that support and repair tissue by producing ECM (e.g., collagen, fibronectin) and usually stay inactive outside of wound healing. Cancer-associated fibroblasts (CAFs) are fibroblast-like cells within the TME that become continuously activated by signals from tumors; they often express markers such as α-SMA, FAP, and PDGFRα/β and can modify the matrix, release cytokines and chemokines, and influence immune responses. In carcinomas, CAFs are generally non-malignant, reactive stromal cells that are clearly different from the epithelial cancer cells in terms of lineage and structure [100].

In sarcomas, many malignant cells themselves exhibit fibroblastic or myofibroblastic programs (e.g., α-SMA, FAP, PDGFRα/β, and vimentin) [101,102,103,104]. This overlap blurs the line between tumor cells and stromal fibroblasts, so marker-based definitions alone can misidentify malignant cells as “CAFs.” Therefore, studies in sarcomas should (i) use lineage/copy-number-aware methods to distinguish aneuploid tumor cells from diploid stromal fibroblasts and (ii) combine markers with functional readouts (matrix stiffening/LOX activity, chemokine gradients such as CXCL12 or CXCL16, and spatial CD8+ T cell access). Accurately making this distinction is essential for interpreting which compartment drives immune exclusion or drug delivery failure and for choosing therapies (e.g., ECM normalization, axis blockade, or payload strategies when stromal antigens are co-expressed with tumor antigens).

In line with the need for lineage-aware and function-first analyses, EwS is genetically and histologically uniform; its tumor cells exhibit varying levels of mesenchymal differentiation at the transcriptional level [105]. By applying single-cell proteogenomic sequencing of EwS cell lines and integrating this data with patients’ tumor transcriptomic data, the researchers identified distinct tumor cell subpopulations, notably a subset marked by a high expression of Cluster of Differentiation 73 (CD73) [105]. CD73+ EwS cells displayed features commonly associated with CAFs, such as the increased expression of ECM-related genes, enhanced migratory capacity, and reduced activity of the EWS: FLI1 fusion oncoprotein [105]. These cells produced large amounts of ECM proteins and were thus called “CAF-like tumor cells.” Spatial profiling revealed that these cells are unevenly distributed throughout tumors, often clustering at invasive fronts and peri-necrotic regions [105]. The findings show that EwS tumor cells can develop CAF-like traits and contribute to remodeling the tumor microenvironment from within. This challenges the traditional view that ECM production in tumors is solely performed by stromal fibroblasts, highlighting an intrinsic mechanism by which tumor cells can influence their microenvironment.

Despite methodological heterogeneity, three themes recur. (1) In osteosarcoma, LOX-high stromal programs track with a stiffened ECM, M2-like macrophage polarization, and pro-invasive niches; pharmacologic LOX inhibition reverses invasive phenotypes and increases tumor cell death in preclinical models. (2) In soft-tissue sarcomas, glycolytic CAFs (GLUT1-dependent) secrete CXCL16, which traps CD8+ T cells at the margins and limits treatment penetration; dampening glycolysis reduces the CXCL16 barrier, restores intratumoral T cell trafficking, and enhances responses to chemotherapy (and ICB). (3) In Ewing sarcoma, CAF-like tumor cells deposit ECM and carry “CAF” transcriptional signatures, indicating that some ostensibly stromal signals arise within malignant clones; lineage-aware analytics are therefore essential before assigning stromal origin. Together with the relative CAF independence of RMS, these patterns clarify when stromal targeting (matrix or metabolism) is plausible and when tumor-intrinsic pathways are likely to dominate.

Despite the ambiguity surrounding cellular identity, accumulating evidence suggests that authentic, non-malignant CAF populations exist in sarcomas and play functional roles in tumor progression, immune modulation, and therapy resistance. These CAFs may originate from resident fibroblasts, pericytes, or bone marrow-derived mesenchymal stem cells (MSCs), like their counterparts in epithelial tumors [11]. A scRNA-seq study of tumors in osteosarcoma explored the TME of primary, recurrent, and metastatic osteosarcoma (OS). The analysis revealed that CAFs are more abundant in recurrent OS and are strongly enriched in the EMT pathway [106]. Notably, CAFs in recurrent tumors showed a high expression of lysyl oxidase (LOX), an enzyme involved in extracellular matrix remodeling and EMT induction. Functional experiments have demonstrated that LOX plays a crucial role in regulating CAF activity, promoting macrophage polarization, and shaping the immune microenvironment in OS [106].

Furthermore, pharmacological inhibition of LOX significantly reduced tumor cell migration and enhanced apoptosis, both in vitro and in vivo. The findings highlight a novel mechanism by which CAFs contribute to OS progression through lysyl oxidase (LOX)-mediated EMT and immune modulation [106]. Targeting LOX in CAFs may offer a promising strategy for remodeling the TME and improving outcomes in recurrent osteosarcoma. While the specific CAF subtypes, known as iCAFs, myCAFs, and apCAFs, have been well characterized in epithelial cancers, such as pancreatic cancer, their classification in sarcomas remains under investigation, and similar but not identical subpopulations have been reported.

Several studies have highlighted key molecular pathways through which CAFs facilitate tumor invasion. A study found that the Collagen Type VI Alpha 1 Chain (COL6A1) is commonly upregulated in OS, especially in lung metastases, and is associated with a poor prognosis [107]. This upregulation is driven by c-Jun binding to the E1A Binding Protein p300 (p300), which increases histone H3 lysine 27 (H3K27) acetylation at the COL6A1 promoter [107]. COL6A1 promotes OS cell migration and invasion by interacting with the Suppressor of Cytokine Signaling 5 (SOCS5) to suppress STAT1 expression and activation via ubiquitination and proteasomal degradation [107]. Additionally, COL6A1 is packaged into osteosarcoma-derived exosomes, which convert normal fibroblasts into cancer-associated fibroblasts (CAFs) that secrete IL-6 and Interleukin-8 (IL-8) [107]. These activated CAFs enhance the invasion and migration of osteosarcoma cells through the TGF-β/COL6A1 signaling pathway [107]. Overall, COL6A1 promotes OS metastasis by both suppressing STAT1 in tumor cells and activating CAFs [107].

Another study identified a specific group of glycolytic CAFs (glyCAFs) in soft tissue sarcomas, characterized by high GLUT1-dependent glycolysis and the expression of CD73 and CD90 [108]. These glyCAFs produce the C-X-C motif chemokine ligand 16 (CXCL16), which traps cytotoxic CD8+ T cells at the tumor margins, restricting their infiltration and promoting an immunosuppressive environment [108]. Inhibiting glycolysis decreased glyCAF accumulation and CXCL16 secretion, improved T cell infiltration, and enhanced chemotherapy response [108]. This study highlights the role of the metabolic state of CAFs in sarcomas in contributing to therapy resistance and immune evasion. These CAFs depend on GLUT1-mediated glycolysis to produce CXCL16, forming a barrier to CD8+ T cell infiltration [108]. Targeting their metabolism restored T cell infiltration and worked synergistically with chemotherapy, revealing a promising strategy to overcome immune exclusion.

Clinical activity, safety, and comparative efficacy: Across modalities, single-agent stromal depletion has yielded limited, context-dependent activity and can be unsafe when architecture-preserving myofibroblastic programs are removed. In contrast, stromal reprogramming (e.g., vitamin A/D analogs, NOX4 inhibition) and axis-directed approaches (e.g., CXCL12/CXCR4, TGF-β, and selected stromal checkpoints) appear more promising in combinations that aim to reopen lymphocyte trafficking, normalize matrix/perfusion, or dismantle chemokine/metabolic barriers rather than ablate the stroma wholesale. Payload-delivery strategies (e.g., antibody–drug conjugates to LRRC15/FAP) can leverage stromal–tumor co-expression but require the demonstration of tumor-dominant expression to mitigate off-tumor toxicity. Safety considerations are non-trivial: A TGF-β blockade may unmask inflammatory toxicities; metabolic inhibition (e.g., glycolysis/GLUT1) carries potential collateral effects on activated T cells; broad FAP targeting may impair wound healing; and aggressive ECM normalization can destabilize vasculature. Comparative efficacy is therefore context-contingent: In LOX-high, stiff, hypoperfused niches, ECM normalization plus chemotherapy and/or ICB is rational; in glycolytic/CXCL16-rich niches, metabolic modulation plus chemotherapy and/or ICB is preferred; where stromal–tumor antigen co-expression is tumor-dominant, payload delivery is attractive; and in CAF-sparse settings (e.g., RMS), tumor-intrinsic targeting should predominate. Prospective studies should pre-specify spatial pharmacodynamic endpoints—such as the intratumoral-to-margin CD8+ T cell ratio, perfusion/hypoxia surrogates, measures of stiffness or LOX activity, and CXCL12/CXCL16 gradients—alongside clinical outcomes to enable a precise benefit–risk assessment and to lay the groundwork for future quantitative synthesis.

Another study showed that reprogramming CAFs, rather than removing them, could be therapeutically beneficial. Their nanocomposite hydrogel delivered an NADPH oxidase 4 (Nox4) inhibitor followed by doxorubicin, reprogramming the stromal environment and boosting Anti-Programmed Cell Death Protein 1 (PD–1) responses in osteosarcoma models [109]. An analysis of 133 soft-tissue sarcoma cases found that high levels of CAF markers, including Fibroblast Activation Protein (FAP), Cluster of Differentiation 10 (CD10), and podoplanin, both within tumors and at their edges, were associated with significantly worse disease-free, metastasis-free, and local recurrence-free survival [110]. These findings underscore the prognostic significance of CAF marker expression in soft-tissue sarcomas, supporting the targeting of the CAF population as a potential therapeutic strategy.

In contrast to other sarcomas, rhabdomyosarcoma (RMS) appears to be largely CAF-independent. Rhabdomyosarcoma (RMS) seems to rely less on stromal fibroblasts. In comparative analyses using 2D and 3D culture systems and mouse xenografts, RMS cells exhibited minimal interaction with fibroblasts, tended to lose their spheroid structure on stromal layers, and showed low CAF infiltration in vivo [111]. These findings indicate that RMS cells directly remodel their ECM and depend less on CAF support for tumor expansion [111]. Analysis of Leucine-Rich Repeat Containing 15 (LRRC15) expression across 711 soft-tissue sarcoma (STS) cases revealed that, unlike in epithelial tumors, LRRC15 is present not only in stromal cells but also in tumor cells in various STS subtypes [112]. Elevated LRRC15 expression was associated with a higher tumor grade and poorer prognosis [28]. Preclinical data demonstrated that targeting LRRC15 with the antibody–drug conjugate ABBV-085 produced substantial anti-tumor effects in LRRC15-positive soft-tissue sarcoma (STS) models [112]. These findings highlight LRRC15 as a potential biomarker and therapeutic target in STS, supporting the ongoing clinical evaluation of ABBV-085. In a first-in-human phase I study of the LRRC15-directed ADC ABBV-085 (NCT02565758), dosing every 14 days identified 3.6 mg/kg as the recommended expansion dose, with manageable safety (fatigue, nausea, and decreased appetite) and a 20% objective response rate in the osteosarcoma/undifferentiated pleomorphic sarcoma cohort (four confirmed partial responses); no monotherapy responses were observed at this dose in other solid tumors, and one partial response occurred with ABBV-085 plus gemcitabine [113]. These data support payload-delivery strategies in LRRC15-positive STS, while underscoring the need for biomarker-gated selection and careful off-tumor risk assessment, particularly when LRRC15 is expressed both in the stromal and tumor compartments.

Two sarcoma-specific pitfalls warrant explicit caution. First, marker overlap (α-SMA, FAP, PDGFRα/β, and vimentin) means that stromal antigens may be co-expressed by tumor cells; depletion strategies risk on-target, pro-tumor shifts if protective stromal elements are removed, while payload strategies (e.g., antibody–drug conjugates) can be advantageous if the expression is tumor-dominant. Second, spatial heterogeneity is pronounced: stromal metabolic and stiffness gradients vary across the tumor core, invasive fronts, and post-therapy beds, so sampling and readouts should be spatially resolved.

In sarcomas, a function-first readout should guide therapy selection. When the stroma shows LOX-high, stiff, hypoperfused niches, pair ECM normalization with cytotoxic chemotherapy or an immune-checkpoint blockade (ICB) to improve perfusion and antigen access. When glycolytic stromal states dominate and CXCL16 creates a chemokine barrier, pursue metabolic modulation to reopen T cell trafficking and potentiate chemotherapy or ICB. Where target antigens are co-expressed by the stroma and tumor (e.g., LRRC15, FAP), prefer payload-delivery strategies if the expression is tumor-dominant, and avoid indiscriminate stromal depletion when the lineage is ambiguous. Finally, in CAF-sparse settings, such as rhabdomyosarcoma, prioritize tumor-intrinsic pathways, as stromal modulation is unlikely to be the main driver.

The evidence base for sarcoma on CAFs remains limited and heterogeneous, encompassing models, assays, and endpoints. Rather than over-interpreting single studies, we synthesize direction-of-effect signals across the available literature (e.g., changes in T cell access, matrix stiffness, perfusion, and invasion surrogates), and we qualify each theme by plausibility and reproducibility across tumor types and platforms. Where quantitative effect sizes were not comparable, we apply a synthesis-without-meta-analysis logic—highlighting consistent patterns and flagging uncertainties—so that conclusions are explicitly hypothesis-generating and depend on future validation.

## 6. Clinical Implications

CAFs are increasingly recognized as pivotal players in shaping the TME and influencing therapeutic outcomes. Their roles in promoting tumor progression, immunosuppression, and resistance to multiple treatment modalities, particularly immunotherapy, make them attractive but complex clinical targets. Emerging therapeutic strategies seek to either eliminate CAFs, reprogram their phenotype, or disrupt their interactions with cancer and immune cells. These approaches must carefully navigate their heterogeneity and context dependence. For instance, the depletion of α-SMA+ myofibroblastic CAFs in PDAC models paradoxically accelerated tumor progression, reduced overall survival, and increased immunosuppressive cell infiltration [113]. These findings underscore the risk of indiscriminate CAF ablation and have shifted therapeutic interest toward selective modulation or the reprogramming of specific CAF subtypes.

Before detailing modality-specific approaches, we outline translational constraints that shape target selection and trial design. Several predictable pitfalls constrain the clinical translation of CAF targeting. First, plasticity and context dependence mean that CAF states interconvert under therapy; depleting or blocking one program can be compensated for by another or by tumor-intrinsic pathways, especially in sarcomas where lineage overlap blurs stromal–tumor boundaries. Second, marker non-specificity (e.g., α-SMA, FAP, and PDGFRα/β) risks on-target activity in non-malignant compartments and the misclassification of malignant mesenchymal cells as “CAFs,” complicating target selection and readouts. Third, spatial heterogeneity (juxtatumoral vs. distal, core vs. invasive front, pre- vs. post-treatment) and sampling biases in dissociation-based assays obscure which stromal niche actually drives immune exclusion or drug delivery failure. Finally, pharmacodynamic measurement in fibrotic, hypoperfused tissue is non-trivial: without built-in spatial biomarkers (e.g., T cell access, perfusion, stiffness, and chemokine gradients), it is difficult to know whether an intervention hit the intended stromal axis. These constraints argue for lineage-aware, spatially resolved, function-first enrichment in trials and for adaptive designs that can pivot when stromal programs shift.

Reprogramming strategies aim to convert tumor-promoting CAFs into quiescent or even tumor-restraining phenotypes. This alternative therapeutic approach focuses on transforming activated CAFs into less harmful fibroblasts or into CAF subpopulations with tumor-suppressive functions rather than eliminating them [114]. One promising agent in this context is all-trans retinoic acid (ATRA), a metabolite of vitamin A known to promote cellular differentiation and immune modulation [115]. ATRA has been shown to restore quiescence in activated fibroblasts, thereby limiting their tumor-supporting behavior in several epithelial tumor models [115].

Beyond its effect on CAFs, ATRA also targets Myeloid-Derived Suppressor Cells (MDSCs), a key immunosuppressive population within the tumor microenvironment [116]. When combined with immune checkpoint inhibitors, ATRA has been shown in preclinical models of mesothelioma, fibrosarcoma, and NSCLC to reduce MDSC accumulation and promote an interferon-driven, CD8+ T cell-enriched tumor milieu, ultimately enhancing the response to anti-PD-1 therapy [116]. These dual actions make ATRA a compelling agent for reshaping the tumor microenvironment and improving immunotherapeutic efficacy. This effect is also supported by phase II trials in melanoma that combine ATRA with ipilimumab or pembrolizumab [117]. Another strategy to limit CAF activity is to restore their quiescent state using retinoic acid (a metabolite of vitamin A) or vitamin D compounds. Vitamin A or D deficiency is linked to CAF activation, so reintroducing these pathways may reverse this effect [118,119]. In PDAC models, ATRA treatment induces CAF quiescence, thereby reducing tumor cell proliferation and increasing apoptosis [120]. Similarly, the vitamin D analog calcipotriol reprograms the stroma to a quiescent state, thereby improving drug delivery and enhancing the efficacy of chemotherapy [121]. Another study found that Minichromosome Maintenance Complex Component 2 (MCM2) is upregulated in liposarcoma tissues and cells, promoting a CAF-like phenotype characterized by an increased expression of FAP, α-SMA, and elevated secretion of IL-6, IL-8, and TGF-β [122]. These MCM2-activated CAF-like cells enhanced liposarcoma proliferation, migration, and invasion. While doxorubicin alone had little effect, combining MCM2 knockdown with doxorubicin suppressed proliferation and induced apoptosis [122]. In vivo, silencing MCM2 reduced tumor growth [122]. Overall, MCM2 drives CAF-like activation and contributes to liposarcoma progression and chemoresistance, suggesting it as a potential therapeutic target [122]. In osteosarcoma, for instance, CAF-derived exosomal microRNA-1228 (miR-1228) promotes tumor invasiveness by suppressing SCAI in cancer cells [123]. Interruption of this axis reduces the metastatic potential and may restore chemosensitivity [123]. Similarly, glycolytic CAFs (glyCAFs) in soft tissue sarcomas create an immunosuppressive niche by producing CXCL16, which establishes a physical and chemotactic barrier to CD8+ T cell infiltration [123]. Targeting glycolysis in these CAFs restored immune cell access and synergized with chemotherapy, offering a new angle for combinatorial therapy.

Despite compelling biology, several context-dependent features of CAF programs complicate clinical translation. Plasticity and state switching (e.g., iCAF, myCAF, and therapy-induced metabolic states) can weaken single-axis interventions over time. Spatial heterogeneity—juxtatumoral versus distal stroma and TLS-proximal versus immune-excluded regions—means that bulk readouts may misclassify the overall effect, undermining patient selection and pharmacodynamic (PD) interpretation. Marker non-specificity (FAP, α-SMA, and PDGFRα/β) creates an on-target/off-tumor risk and, in sarcomas, can confuse stromal fibroblasts with malignant mesenchymal cells. Matrix mechanics and perfusion barriers further restrict drug delivery, making it unclear whether a lack of efficacy reflects target failure or inadequate exposure. Finally, redundant stromal checkpoints (e.g., TNC/integrins, Endo180) can allow compensatory escape when single nodes are blocked. Overcoming the issue requires trials that pre-define stromal context strata (spatial/immune/matrix), use biomarker-based eligibility, include on-treatment PD readouts (stromal cues, spatial T cell metrics, and stiffness/LOX activity), and favor combination strategies aligned with the dominant CAF program rather than uniform “anti-CAF” depletion methods.

Across major strategy classes, clinical and early-phase data are varied but indicate consistent signals that fit the context. Reprogramming and normalization (such as ATRA, vitamin D analogs, and NOX4 inhibition) promote stromal quiescence and enhance drug access in desmoplastic, architecture-preserving niches; safety is generally manageable but requires metabolic and endocrine monitoring. Axis blockades, such as CXCL12/CXCR4 or TGF-β, combined with ICB, can enhance immune trafficking and reverse exclusion in IL-1/iCAF-dominant environments. However, these approaches have class-specific toxicities that necessitate phased escalation and biomarker guidance. Targeting glycolytic CAFs (e.g., GLUT1-linked pathways) metabolically can restore infiltration and chemosensitivity in niches rich in glycolytic activity and CXCL16. ECM normalization (e.g., LOX inhibition, TNC/integrin interference) improves perfusion and access to antigens and drugs in stiff, poorly perfused tumors but may carry wound-healing or vascular risks that require dose and schedule optimization. Conversely, the indiscriminate depletion of myCAF can be detrimental in α-SMA-rich desmoplasia, such as in PDAC, emphasizing the importance of aligning interventions with the stromal context rather than pursuing broad “anti-CAF” strategies. Head-to-head data are limited; therefore, we focus on the consistency of benefits within the relevant context rather than relying solely on absolute response rates.

Trials should prospectively stratify by stromal program (myCAF-rich/desmoplastic; iCAF/immune-excluded; glycolytic/meCAF; and checkpoint-high stroma), incorporate adaptive elements, and co-report safety and PD evidence of stromal change alongside clinical endpoints.

### 6.1. CAF Subtypes as Predictive Biomarkers

Beyond their role as therapeutic targets, CAFs and their molecular signatures are emerging as potential predictive biomarkers. Single-cell transcriptomic analyses have identified subtypes of CAFs that correlate with immune exclusion, desmoplasia, and poor responses to immunotherapy. For instance, in NSCLC, CAF-based gene expression profiles have been shown to predict resistance to immune checkpoint blockades [124]. Across tumors, iCAFs and myCAFs mark the opposite ends of an inflammatory/secretory versus contractile/ECM remodeling continuum. Spatially, iCAFs are often enriched in regions more distal to tumor nests, whereas myCAFs are typically juxtatumoral; however, this gradient is context- and time-dependent rather than absolute [125]. Both states contribute substantially to extracellular matrix production and remodeling, and the resulting desmoplastic stroma—particularly the dense, stiff ECM shaped by myofibroblastic programs—functions as a physical and biochemical barrier to CD8+ T cell trafficking, reinforcing immune exclusion [125,126]. Additionally, ECM-rich myCAF signatures have been correlated with increased T cell dysfunction, as indicated by the elevated expression of checkpoint molecules, such as PD-1 and CTLA-4, on tumor-infiltrating lymphocytes [72].

### 6.2. CAFs and Therapeutic Resistance

CAFs play a crucial role in promoting resistance to cancer treatments by supporting the survival of tumor cells under therapeutic stress. CAFs also remodel the ECM, creating a dense, fibrotic microenvironment that impedes drug penetration and promotes tumor cell invasion [127,128,129]. Additionally, CAFs facilitate immune evasion by attracting immunosuppressive cells (e.g., MDSCs, Tregs) and preventing the infiltration of cytotoxic T cells, further supporting tumor persistence [130,131,132]. Overall, these complex interactions position CAFs as key orchestrators of treatment resistance, making them potential targets for therapy to improve clinical outcomes. These features enable tumors to evade cytotoxic effects and adapt to the pressures of treatment.

#### 6.2.1. Chemotherapy Resistance

CAFs are increasingly recognized as central mediators of chemotherapy resistance in various solid tumors (Figure 2). Through a combination of paracrine signaling, extracellular vesicle (EV) release, and metabolic reprogramming, CAFs reshape the tumor microenvironment to shield cancer cells from drug-induced cytotoxicity [133,134]. A consistent finding across malignancies is the role of CAF-derived cytokines, particularly Interleukin-6 (IL-6) [135]. IL-6 secretion activates the STAT3 signaling pathway in tumor cells, promoting survival, proliferation, and resistance to Deoxyribonucleic Acid (DNA)-damaging agents, such as cisplatin [136]. In esophageal squamous cell carcinoma, IL-6 collaborates with exosomal miR-21 to induce monocytic myeloid-derived suppressor cells (M-MDSCs), which further contribute to immunosuppression and chemoresistance [137]. Inhibiting both IL-6 and miR-21 partially reversed resistance, highlighting the therapeutic relevance of this axis [138].

Another mechanism by which CAFs promote chemoresistance is through the transfer of non-coding RNAs via exosomes [139]. In UC, exosomal Long Intergenic Non-Protein Coding RNA 355 (LINC00355) derived from CAFs was shown to sponge microRNA-34b-5p (miR-34b-5p), consequently upregulating the ATP-Binding Cassette Subfamily B Member 1 (ABCB1), a key drug efflux transporter [140]. This mechanism directly reduced cisplatin sensitivity in tumor cells and was reversed upon knockdown of LINC00355 or the overexpression of miR-34b-5p [140]. Similarly, in ovarian and oral cancers, CAF-secreted midkine (MK) was found to upregulate the long non-coding RNA Antisense Non-Coding RNA in the INK4 Locus (ANRIL) in cancer cells [141]. ANRIL modulates apoptosis regulators and efflux transporters, ultimately reducing the efficacy of cisplatin [141]. These findings demonstrate that long non-coding RNA (lncRNA)-mediated crosstalk between CAFs and tumor cells significantly contributes to therapy resistance.

In addition to transcriptional changes, CAFs also rewire the metabolism of cancer cells to favor survival under chemotherapeutic stress. In ovarian cancer, the CAF-derived Holliday Junction Recognition Protein (HJURP) enhances glutamine metabolism and tricarboxylic acid (TCA) cycle activity, facilitating resistance to doxorubicin [142]. This metabolic adaptation not only sustains energy production but also buffers redox imbalances caused by chemotherapy. Importantly, the co-culture of tumor cells with CAF-conditioned media results in an elevated HJURP expression and increased IC50 values, highlighting the functional relevance of this stromal–tumor metabolic axis [142].

#### 6.2.2. CAFs and Resistance to Immunotherapy

CAFs have emerged as crucial mediators of resistance to immune checkpoint blockades (ICBs) across various tumor types, and in some tumor types, PD-L1/PD-L2 expressions on subsets of CAFs are reported, which are features that may contribute to immune dysfunction (Figure 2). The reported prevalence varies by tumor type [72,74,143]. Studies have demonstrated that targeting this axis—through the inhibition of C-X-C Chemokine Receptor Type 4 (CXCR4) or TGF-β signaling—can remodel the tumor stroma, enhance CD8+ T cell infiltration, and improve the effectiveness of immune checkpoint blockades [144,145]. Recent studies have improved our understanding of CAF heterogeneity in this context. For instance, subsets such as iCAFs and myCAFs are typically immunosuppressive, whereas apCAFs may support anti-tumor responses [146,147]. An early phase II, open-label, multicenter study evaluated the unconjugated anti-FAP monoclonal antibody sibrotuzumab (BIBH1) in metastatic CRC. Twenty-five patients received 100 mg IV weekly for up to 12 weeks. Efficacy was minimal: among 17 evaluable patients (≥8 doses), no objective responses were seen and only two had transient stable disease; all others progressed. The drug was generally well tolerated (rigors/chills, nausea, and flushing; one bronchospasm), with anti-drug antibodies in ~12.5% of patients receiving ≥2 infusions. These experiences underscore the need for biomarker-gated eligibility, combination designs (e.g., CXCR4 blockade to disrupt CXCL12-mediated exclusion), and careful on-target/off-tumor safety monitoring when engaging broadly expressed stromal markers. Clinical correlations are emerging; Microfibril-Associated Glycoprotein 2 (MFAP2) plus CAFs in GC have been shown to impair responses to both chemotherapy and immunotherapy through macrophage migration inhibitory factor (MIF)-mediated immune modulation [148]. Similarly, DCN-rich desmoplastic signatures have been reported to be associated with ICB-resistant phenotypes in metastatic GC [149]. Mechanistically, CAFs can induce resistance through direct interactions with immune cells or by remodeling the extracellular matrix. Endo180 (MRC2), a receptor enriched on myCAFs, has been shown to suppress CD8+ T cell infiltration and limit the efficacy of αPD-1/αCTLA-4 in murine models and melanoma patients [150]. In PDAC, CAF-secreted Tenascin C (TNC) promotes epithelial-to-mesenchymal transition (EMT) and immune exclusion through integrin αV/β3 signaling, identifying a novel stromal checkpoint amenable to therapeutic targeting [151]. Emerging resistance mechanisms are also evident in cellular therapies. In multiple myeloma, CAFs within the bone marrow niche decreased Chimeric Antigen Receptor T cell therapy (CAR-T) efficacy by impairing cytotoxic T cell function [152]. Dual targeting of CAFs and tumor cells restored the response, indicating the need to design CAR-T constructs that address stromal barriers.

FAP^+^ stromal cells have been shown to suppress anti-tumor immunity [153], and CXCL12 produced by FAP^+^ CAFs drives T cell exclusion and blunts responses to αCTLA-4/αPD-L1, findings that motivated later CXCR4-blockade combinations [154]. Seminal mouse work established that FAP^+^ stromal cells form a non-redundant, immunosuppressive compartment: genetic ablation of FAP^+^ stromal cells (~2% of the tumor) restored immune control of tumor growth via IFN-γ/TNF-α/TNF-α-dependent mechanisms and induced rapid hypoxic necrosis of tumor and stromal cells [153]. This supports viewing CAF programs as “stromal checkpoints” and provides a mechanistic rationale for stromal-targeted combinations rather than indiscriminate depletion. In pancreatic ductal adenocarcinoma (PDAC), CXCL12 produced by FAP^+^ CAFs coats tumor cells and excludes T cells; CXCR4 blockade (AMD3100) disrupts this barrier, permits intratumoral CD8^+^ accumulation, and synergizes with anti-PD-L1 [154]. These data mechanistically link iCAF-like chemokine programs to immune exclusion and motivate biomarker-guided CXCR4/ICB combinations in iCAF-dominant contexts. Also, Cancer-associated fibroblasts (CAFs) can suppress anti-tumor immunity by rewiring T-cell metabolism, chiefly through Indoleamine 2,3-dioxygenase (IDO) and Arginase 2 (ARG2). IDO depletes tryptophan and generates kynurenine, creating metabolic starvation that halts T-cell proliferation, triggers apoptosis or arrest, and skews differentiation toward immunosuppressive Tregs. In parallel, ARG2 consumes arginine, undermining TCR signaling (including CD3 expression) and driving T-cell dysfunction and suppressive phenotypes. Clinically, ARG2^+^ CAFs in PDAC localize to hypoxic zones, correlate with fewer CD4^+^/CD8^+^ infiltrates, and independently predict worse survival. Therapeutically, blocking these pathways could restore T-cell function and enhance the efficacy of immunotherapy. Still, the TME’s metabolic network is plastic, so single-node inhibition risks compensation, arguing for rational combinations and context-aware targeting.

Finally, CAF-driven resistance extends beyond immune exclusion. Recent studies suggest that CAFs may also foster resistance by inhibiting ferroptosis in tumor cells—a type of regulated cell death that can enhance anti-tumor immune responses through the release of danger-associated molecular patterns (DAMPs) and the promotion of immune activation [155,156]. For instance, in gastrointestinal tumors, the overexpression of ANO1 facilitates CAF recruitment via TGF-β signaling, resulting in the formation of an immunosuppressive niche that resists anti-PD-1 therapy [157]. Anoctamin-1 (ANO1) has also been linked to the modulation of lipid metabolism and iron homeostasis, which may contribute to ferroptosis resistance [157]. Inhibition of ANO1 reverses both CAF accumulation and ferroptosis suppression, presenting a therapeutic opportunity to enhance immunotherapy responses by targeting CAF-related metabolic and signaling pathways [157].

#### 6.2.3. CAFs and Resistance to Endocrine and Targeted Therapies

CAFs play a complex role in mediating resistance to endocrine and targeted therapies across different cancer types (Figure 3). In hepatocellular carcinoma (HCC), researchers developed murine and human 3D co-culture models of liver tumor organoids with CAFs to investigate their interactions and treatment responses [158]. CAFs promoted tumor organoid growth through direct contact and paracrine signaling, while cancer cells influenced the physiology of CAFs. In xenograft models, co-transplantation with CAFs has been shown to enhance tumor growth [158]. Additionally, the presence of CAFs or their conditioned medium reduced the effectiveness of sorafenib, regorafenib, and 5-Fluorouracil (5-FU), underscoring the role of CAFs in fostering liver cancer growth and therapy resistance [156]. Also in HCC, another study found that fibronectin extra domain A (FN-EDA), derived from cancer-associated fibroblasts, plays a key role in promoting sorafenib resistance in hepatocellular carcinoma (HCC) [159]. FN-EDA activates the Toll-Like Receptor 4/Nuclear Factor kappa-light-chain-enhancer of activated B cells (TLR4/NF-κB) pathway in HCC cells, leading to the increased expression of Serine Hydroxymethyltransferase 1(SHMT1), a crucial enzyme in one-carbon metabolism, which helps cancer cells counter sorafenib-induced oxidative stress [159]. In NSCLC, cancer-associated fibroblasts (CAFs) drive resistance to EGFR-TKIs through multiple mechanisms. Yi et al. demonstrated that CAF-secreted HGF and IGF-1 activate the Mesenchymal–Epithelial Transition factor/Insulin-like Growth Factor 1 Receptor (c-MET/IGF-1R) signaling pathway, inducing Annexin A2 (ANXA2) expression and EMT, which mediates gefitinib resistance [160]. Zhang et al. described a subset of Collagen Triple Helix Repeat Containing (CTHRC1+) CAFs that sustain EGFR-TKI resistance through metabolic reprogramming and histone acetylation [161]. Clinically, high stromal ANXA2 and CAF density were correlated with poor progression-free survival [162]. Podoplanin-positive CAFs have been shown to promote primary resistance in tumors by sustaining persistent ERK pathway activation, which supports the survival and proliferation of tumor cells despite therapy [9]. Meanwhile, C-X-C motif chemokine ligand 14 (CXCL14)-positive CAFs contribute to cancer progression by enhancing epithelial–mesenchymal transition (EMT) and promoting angiogenesis, thereby facilitating increased tumor invasiveness and vascularization [163]. In breast cancer, one study identified a subset of Cluster of Differentiation 63 (CD63^+^) CAFs in ER-positive breast cancer that are enriched in tumors resistant to CDK4/6 inhibitors (CDK4/6i) [164]. These CAFs promote resistance by releasing exosomes containing miR-20, which targets and downregulates Retinoblastoma 1 (RB1) in cancer cells, thereby reducing the sensitivity to Cyclin-Dependent Kinases 4 and 6 (CDK4/6) inhibitors, using cyclic Arg-Gly-Asp peptide (cRGD)-miR-20 sponge nanoparticles to neutralize the miR-20 restored drug sensitivity in vitro and in vivo, suggesting that targeting CD63^+^ CAFs and their exosomal miR-20 could improve CDK4/6i efficacy in breast cancer [162]. Another study identified a subset of CD63^+^ CAFs in Estrogen Receptor Alpha (Erα)-positive breast cancer that contributes to tamoxifen resistance [136]. These CAFs secrete exosomes enriched with miR-22, which targets the estrogen receptor α (Erα) and the Phosphatase and Tensin Homolog (PTEN) in cancer cells, promoting drug resistance [165]. The Splicing Factor Arginine/Serine-Rich 1 (SFRS1) mediates packaging of miR-22 into exosomes, while CD63 supports the CAF phenotype via STAT3 activation [166]. Blocking CD63^+^ CAFs with neutralizing antibodies or miR-22 sponge nanoparticles restored tamoxifen sensitivity, highlighting CD63^+^ CAFs as potential therapeutic targets to overcome resistance [166].

Also in breast cancer, CAFs influence responses to Anti-Human Epidermal Growth Factor Receptor 2 (HER2) therapies (e.g., trastuzumab) by increasing cancer stem cell populations and activating IL–6-mediated Signal Transducer and Activator of Transcription 3/Phosphoinositide 3-Kinase (STAT3/PI3K) pathways [167]. Additionally, low stromal Extracellular Signal-Regulated Kinase (ERK) phosphorylation predicted a poor response to tamoxifen, and the loss of caveolin-1 in cancer-associated fibroblasts (CAFs) was associated with worse outcomes in hormonal therapy [168]. In prostate cancer, CAF-secreted C-C Motif Chemokine Ligand 5 (CCL5) increases Androgen Receptor (AR) and PD-L1 expression, promoting resistance to enzalutamide [169], and CTHRC1+ myofibroblast-like CAFs modulate Androgen Receptor signaling through the Cellular Communication Network Factor 2/caveolin-1/Androgen Receptor (CCN2/CAV1/AR) pathway, further contributing to anti-androgen resistance [170]. In ovarian cancer, CAFs mediate resistance to anti-angiogenic therapy through sustained pro-angiogenic signaling and the expression of immune checkpoints [171]. Similarly, in head and neck squamous cell carcinoma (HNSCC), CAFs promote resistance to EGFR inhibitors through the secretion of Matrix Metalloproteinases (MMPs) and Amphiregulin (AREG)-mediated receptor stabilization [172]. Neuroendocrine tumors (NETs) show resistance to everolimus due to CAF-induced STAT3 activation, which promotes proliferation and drug resistance [173].

## 7. Challenges and Future Directions

Despite the growing interest in CAF-targeted therapies, clinical translation remains limited due to several unresolved challenges. A significant obstacle is the absence of specific, universal CAF markers that distinguish pro-tumorigenic from tumor-restraining subsets, which increases the risk of off-target effects and disrupts regular tissue repair [87]. Additionally, the inherent plasticity of CAFs allows for dynamic phenotype switching under therapeutic pressure, raising the possibility that targeting one subtype may lead to the compensatory activation of others, thereby preserving tumor-supportive functions [9,97]. These challenges are particularly evident in mesenchymal tumors, such as sarcomas, where the distinction between malignant tumor cells and CAFs is often unclear due to overlapping lineage markers [171]. This complicates efforts to design stromal-specific interventions and highlights the need for improved lineage-tracing and spatial profiling techniques in these tumors. Future research should focus on developing subtype-specific and context-dependent cancer-associated fibroblast (CAF) therapies, guided by single-cell RNA sequencing, spatial transcriptomics, and epigenetic profiling. These technologies may facilitate the identification of actionable CAF subpopulations that drive resistance to treatment or immune exclusion. Longitudinal studies exploring the temporal evolution of CAF phenotypes during chemotherapy, radiotherapy, and immunotherapy will be essential for understanding how CAFs adapt throughout treatment.

Simultaneously, organotypic 3D culture models and patient-derived tumor–stroma co-cultures can act as physiologically relevant systems to assess the effectiveness of CAF-modulating agents. These models may aid in unraveling the reciprocal interactions between cancer-associated fibroblasts (CAFs), cancer stem cells, immune populations, and endothelial cells, which are critical relationships that impact tumor relapse and metastasis. Additionally, incorporating CAF profiling into the design of clinical trials could improve patient stratification, especially in stromally dense or immune-excluded tumors like PDAC, soft tissue sarcomas, and desmoplastic GC. Stromal gene signatures may function not only as prognostic markers but also as predictive tools for combinatorial therapies.

Future studies should pre-specify stromal context strata, such as juxtatumoral versus distal, TLS-proximal versus immune-excluded, and desmoplastic versus glycolytic, and test CAF-modulating strategies within each stratum. Adaptive randomization guided by stromal biomarker panels (CAF metagenes, CXCL12/CXCL16 gradients, ECM stiffness/LOX activity, and spatial CD8+ T cell access) can adjudicate competing hypotheses and curb citation bias. By tying pre-specified spatial/functional pharmacodynamic endpoints to clinical readouts, such designs convert the “CAF paradox” into falsifiable predictions rather than narrative claims.

## 8. Conclusions

Cancer-associated fibroblasts (CAFs) are dynamic and multifaceted components of the tumor microenvironment. Their ability to support tumor growth, suppress anti-tumor immunity, and mediate therapy resistance positions them as key therapeutic targets, while also making them complex, context-dependent regulators. Notably, CAFs can also inhibit tumor progression under certain conditions, underscoring the need for precise, subtype-specific targeting strategies rather than indiscriminate depletion.

This review highlights the diversity, adaptability, and dual roles of CAFs in both epithelial and mesenchymal tumors. While the biology of CAFs in carcinomas has been widely studied, our understanding of CAF-like populations in sarcomas remains limited, further complicated by the overlapping lineage between malignant and stromal cells. Bridging this knowledge gap is crucial for advancing therapeutic strategies in these aggressive and often treatment-resistant cancers.

Future efforts should focus on refining CAF classification, identifying functional subtypes, and integrating stromal profiling into clinical decision-making. Combining CAF-targeted interventions with immunotherapy, chemotherapy, or anti-angiogenic agents may yield synergistic benefits, particularly in stromally dense and immunologically “cold” tumors.

Ultimately, unlocking the therapeutic potential of CAF modulation will require context-aware, personalized approaches informed by a comprehensive understanding of the tumor microenvironment’s spatial and temporal dynamics.

## Figures and Tables

**Figure 1 cells-14-01398-f001:**
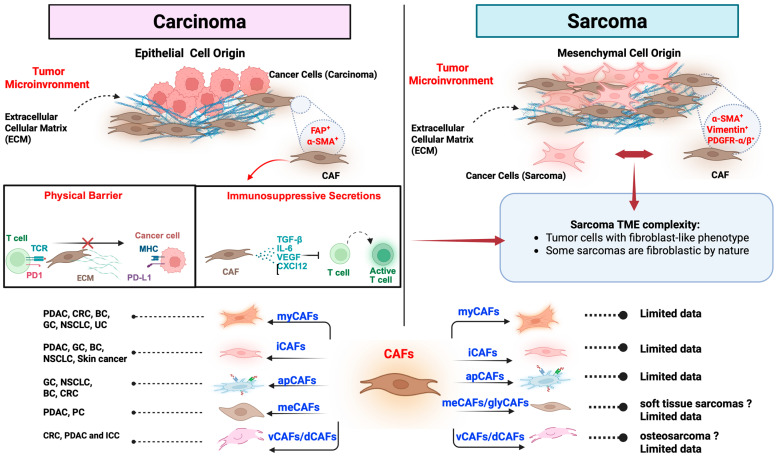
Distinct origins and functional heterogeneity of cancer-associated fibroblasts in carcinomas and sarcomas. The figure presents a detailed comparison of cancer-associated fibroblasts (CAFs) in carcinomas and sarcomas, describing their cellular origins, interactions with the tumor microenvironment (TME), and distribution by subtype. Carcinomas originate from epithelial cells. CAFs in these tumors typically express markers such as FAP^+^ and α-SMA^+^. They influence tumor progression through two primary mechanisms. First, they form a physical barrier via the extracellular matrix (ECM), which restricts T cell infiltration and prevents direct immune recognition of cancer cells. Second, they secrete immunosuppressive factors—such as TGF-β, IL-6, VEGF, and CXCL12—that inhibit T cell activation and sustain an immunosuppressive TME. Within carcinomas, multiple CAF subtypes have been identified: myCAFs (in PDAC, CRC, BC, GC, NSCLC, UC), iCAFs (in PDAC, GC, BC, NSCLC, skin cancer), apCAFs (in GC, NSCLC, BC, CRC), meCAFs (in PDAC, PC), and vCAFs/dCAFs (in CRC, PDAC, ICC). Sarcomas derive from mesenchymal cells and often feature CAFs expressing α-SMA^+^, vimentin^+^, and PDGFR-α/β^+^. Their TME is particularly complex because some sarcoma tumor cells adopt a fibroblast-like phenotype, and certain sarcomas are inherently fibroblastic. CAF subtypes identified in sarcomas include myCAFs, iCAFs, apCAFs, meCAFs/glyCAFs, and vCAFs/dCAFs; however, data on these subtypes is limited—particularly for soft tissue sarcomas and osteosarcomas—indicating a need for further research. PDAC: Pancreatic ductal adenocarcinoma, CRC: Colorectal cancer, BC: Breast cancer, GC: Gastric cancer, NSCLC: Non-small cell lung cancer, UC: Urothelial carcinoma, PC: Prostate cancer, ICC: Intrahepatic cholangiocarcinoma, myCAFs: Myofibroblastic CAFs, iCAFs: Inflammatory CAFs, apCAFs: Antigen-presenting CAFs, meCAFs: Metabolic CAFs, vCAFs: Vascular CAFs, dCAFs: Developmental CAFs, glyCAFs: Glycolytic CAFs.

**Figure 2 cells-14-01398-f002:**
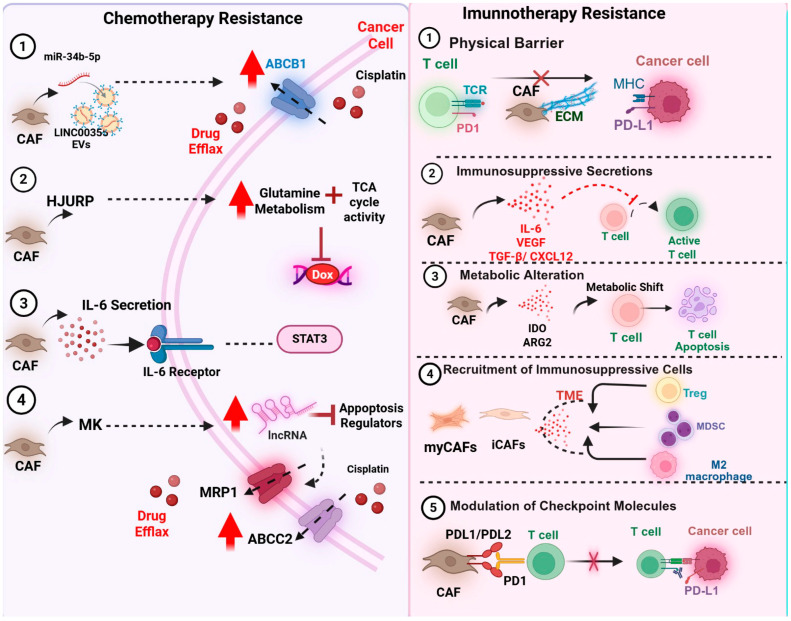
CAF-mediated mechanisms of resistance to chemotherapy and immunotherapy in the tumor microenvironment. The **left** panel illustrates four CAF-driven routes that make cancer cells less sensitive to cytotoxic drugs. (1) microRNA/extracellular-vesicle route: CAFs package miR-34b-5p and the long non-coding RNA LINC00355 into extracellular vesicles (EVs). Delivery of these cargoes to cancer cells upregulates the drug-efflux transporter ABCB1, thereby increasing cisplatin efflux and reducing intracellular drug levels. (2) Metabolic reprogramming via HJURP: CAFs (through signals that elevate HJURP) drive a shift toward higher glutamine metabolism and increased TCA-cycle activity in cancer cells. The enhanced mitochondrial/TCA throughput antagonizes the DNA-damaging action of doxorubicin (Dox) and supports survival under chemotherapy. (3) IL-6 → STAT3 survival signaling: CAFs secrete IL-6, which engages the IL-6 receptor on cancer cells and activates STAT3. STAT3 signaling promotes pro-survival programs and contributes to broad chemoresistance. (4) MK/lncRNA–efflux/anti-apoptosis axis: CAF-derived MK (midkine) induces long non-coding RNAs in cancer cells and upregulates apoptosis regulators together with the efflux pumps MRP1 and ABCC2. The result is greater drug efflux (including cisplatin) and reduced apoptosis. The **right** panel summarizes five CAF-mediated barriers to effective T-cell responses. (1) Physical barrier: Dense ECM deposited and organized by CAFs forms a structural blockade. A T cell (TCR^+^, PD-1^+^) is shown being physically excluded from contacting a cancer cell (expressing MHC and PD-L1). (2) Immunosuppressive secretions: CAFs release soluble mediators—IL-6, VEGF, TGF-β, and CXCL12—that dampen T-cell activation, foster abnormal angiogenesis, and maintain an immunosuppressive milieu. (3) Metabolic alteration of T cells: CAFs express enzymes such as IDO and ARG2 that deplete or reroute key amino acids, driving a metabolic shift in T cells and promoting T-cell apoptosis or dysfunction. (4) Recruitment of immunosuppressive cells: Via chemokines/cytokines emanating from different CAF states (myCAFs, iCAFs), the TME is enriched with Tregs, MDSCs, and M2 macrophages, all of which suppress anti-tumor immunity. (5) Checkpoint modulation: CAFs themselves can express PD-L1/PD-L2, engaging PD-1 on T cells to inhibit their activity, while tumor cells concurrently express PD-L1, compounding checkpoint inhibition. Cancer-Associated Fibroblast (CAF); Extracellular Matrix (ECM); Tumor Microenvironment (TME); microRNA-34b-5p (miR-34b-5p); Long Intergenic Non-coding RNA 00355 (LINC00355); Extracellular Vesicles (EVs); ATP-Binding Cassette Subfamily B Member 1/P-glycoprotein (ABCB1); Cisplatin; Drug Efflux; Holliday Junction Recognition Protein (HJURP); Tricarboxylic Acid cycle (TCA); Doxorubicin (Dox); Interleukin-6 (IL-6); Signal Transducer and Activator of Transcription 3 (STAT3); Midkine (MK); Long Non-coding RNA (lncRNA); Multidrug Resistance-Associated Protein 1 (MRP1/ABCC1); ATP-Binding Cassette Subfamily C Member 2 (ABCC2); T-Cell Receptor (TCR); Programmed Cell Death Protein-1 (PD-1); Major Histocompatibility Complex (MHC); Programmed Death-Ligand 1 (PD-L1); Vascular Endothelial Growth Factor (VEGF); Transforming Growth Factor-beta (TGF-β); C-X-C Motif Chemokine Ligand 12 (CXCL12); Indoleamine 2,3-Dioxygenase (IDO); Arginase-2 (ARG2); Regulatory T cell (Treg); Myeloid-Derived Suppressor Cell (MDSC); M2-polarized macrophage (M2); Programmed Death-Ligand 2 (PD-L2); Myofibroblastic CAFs (myCAFs); Inflammatory CAFs (iCAFs).

**Figure 3 cells-14-01398-f003:**
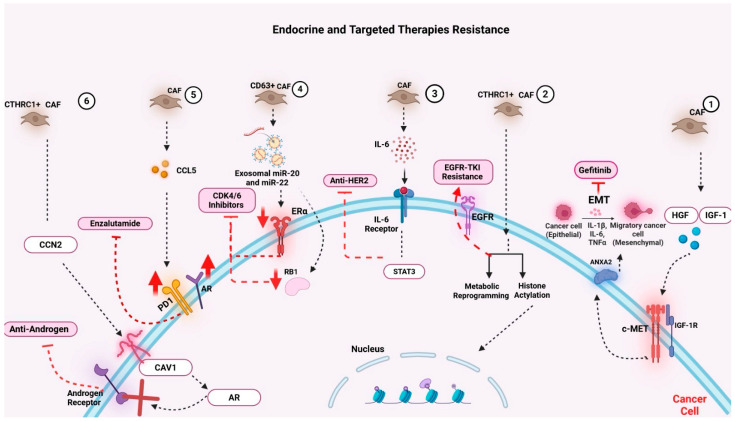
CAF-mediated mechanisms of resistance to endocrine and targeted therapies in cancer. (1) Paracrine growth-factor bypass: A generic CAF secretes HGF and IGF-1, which activate c-MET and IGF-1R on cancer cells. These inputs create a bypass around upstream inhibitors (e.g., EGFR blockade), sustain survival/mitogenic signaling, and facilitate EMT. The cartoon shows an epithelial cell shifting to a migratory mesenchymal-like state, accompanied by inflammatory cytokines (IL-1β, IL-6, TNFα) and ANXA2—features associated with gefitinib resistance. (2) CTHRC1^+^ CAF → EGFR-TKI resistance, A CTHRC1^+^ CAF population is linked to resistance to EGFR tyrosine-kinase inhibitors (EGFR-TKIs). Mechanistically, the panel highlights two downstream layers in cancer cells—metabolic reprogramming and histone acetylation—that together sustain EGFR pathway activity (or compensate for its inhibition), driving therapeutic escape. (3) IL-6 → STAT3 undermines anti-HER2 therapy, Conventional CAFs secrete IL-6, which engages the IL-6 receptor on tumor cells and activates STAT3. This survival transcription program antagonizes anti-HER2 treatments, preserving growth signals despite HER2 targeting. (4) CD63^+^ CAF exosomes blunt ERα / CDK4/6 inhibition, A CD63^+^ CAF subset exports exosomes loaded with miR-20 and miR-22. Delivery of these microRNAs to tumor cells suppresses ERα signaling and perturbs the RB1 node, thereby diminishing the effectiveness of CDK4/6 inhibitors (which rely on an intact RB checkpoint and endocrine dependency). (5) CCL5 amplifies AR–PD1 axes, eroding endocrine control, CAF-derived CCL5 increases androgen-signaling tone and immune-checkpoint pressure. In the schematic, AR activity rises in parallel with PD1/PD-1 axis components, contributing to reduced sensitivity to CDK4/6 inhibitors and endocrine agents by reinforcing pro-survival signaling and immunosuppressive crosstalk. (6) CTHRC1^+^ CAF → CCN2 → anti-androgen resistance. Another CTHRC1^+^ CAF program upregulates CCN2 in cancer cells, which is associated with resistance to anti-androgens such as enzalutamide. The pathway intersects with CAV1 (caveolin-1) and AR trafficking/signaling, supporting androgen-receptor activity despite pharmacologic blockade. Cancer-Associated Fibroblast (CAF); Collagen Triple Helix Repeat Containing 1 (CTHRC1); Connective Tissue Growth Factor (CCN2/CTGF); Chemokine (C-C motif) Ligand 5 (CCL5); Exosome marker (CD63); Epidermal Growth Factor Receptor (EGFR); EGFR Tyrosine-Kinase Inhibitor (EGFR-TKI); Hepatocyte Growth Factor (HGF); Insulin-Like Growth Factor 1 (IGF-1); IGF-1 Receptor (IGF-1R); Signal Transducer and Activator of Transcription 3 (STAT3); Interleukin-6 (IL-6); Interleukin-1 beta (IL-1β); Tumor Necrosis Factor-alpha (TNFα); Epithelial-to-Mesenchymal Transition (EMT); Annexin A2 (ANXA2); c-MET proto-oncogene receptor tyrosine kinase (c-MET); Androgen Receptor (AR); Estrogen Receptor alpha (ERα); Retinoblastoma protein (RB1); Caveolin-1 (CAV1); Programmed cell death protein-1 (PD-1); Programmed death-ligand 1 (PD-L1); Tyrosine-kinase inhibitor (TKI); MicroRNA (miR); Gefitinib (EGFR-TKI).

## Data Availability

No new data were created or analyzed in this study. Data sharing does not apply to this article.

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
