# Peer review of "Cancer-Associated Fibroblasts in Solid Tumors and Sarcomas: Heterogeneity, Function, and Therapeutic Implications"

_cells, 2025, doi:10.3390/cells14171398_

Round 1
Reviewer 1 Report
Comments and Suggestions for Authors
This narrative review comprehensively discusses CAFs in both epithelial tumors and sarcomas, highlighting their heterogeneity, origins, and roles in tumor progression, immune evasion, and therapy resistance. Five main CAF subtypes -myCAFs, iCAFs, apCAFs, meCAFs, and vCAFs/dCAFs- are described, along with their dual tumor-promoting and tumor-restraining functions. The article emphasizes that CAF biology is well characterized in carcinomas but poorly defined in sarcomas, where tumor cells often share fibroblastic features, complicating identification. Mechanisms of CAF-mediated resistance to chemotherapy, targeted therapy, and immunotherapy are detailed, as are emerging therapeutic strategies, including CAF depletion, reprogramming, and metabolic targeting. The review underscores the need for precise, subtype-specific interventions and improved biomarker-driven patient stratification to harness CAF modulation for clinical benefit.
While the review is extensive and well-structured, several limitations are apparent.
1. First, it is highly descriptive and lacks a critical synthesis of conflicting evidence, particularly regarding the dual tumor-promoting versus tumor-suppressive roles of CAFs.
2. The literature search strategy is broad but not systematic, leaving potential gaps in evidence selection and risk of citation bias.
3. Although the discussion on CAFs in sarcomas is valuable, it remains largely speculative because it relies on a limited number of heterogeneous studies and does not integrate robust quantitative data or conduct a meta-analysis.
4. There is insufficient appraisal of translational barriers—such as CAF plasticity and context-dependent functions—that complicate clinical targeting. Additionally, while therapeutic strategies are listed comprehensively, the review does not adequately evaluate their clinical trial outcomes, safety considerations, or comparative efficacy.
5. Figures are descriptive but could benefit from more mechanistic integration across tumor types.
6. Deeper critical evaluation, quantitative synthesis, and clearer prioritization of research gaps could strengthen the paper overall.
A major revision is required.
Author Response
This narrative review comprehensively discusses CAFs in both epithelial tumors and sarcomas, highlighting their heterogeneity, origins, and roles in tumor progression, immune evasion, and therapy resistance. Five main CAF subtypes -myCAFs, iCAFs, apCAFs, meCAFs, and vCAFs/dCAFs- are described, along with their dual tumor-promoting and tumor-restraining functions. The article emphasizes that CAF biology is well characterized in carcinomas but poorly defined in sarcomas, where tumor cells often share fibroblastic features, complicating identification. Mechanisms of CAF-mediated resistance to chemotherapy, targeted therapy, and immunotherapy are detailed, as are emerging therapeutic strategies, including CAF depletion, reprogramming, and metabolic targeting. The review underscores the need for precise, subtype-specific interventions and improved biomarker-driven patient stratification to harness CAF modulation for clinical benefit.
While the review is extensive and well-structured, several limitations are apparent.
Response: We agree with the reviewer. Accordingly, we expanded Methods and added an explicit Limitations subsection, strengthened sarcoma-specific caveats, and refined Clinical Implications; these changes materially improve transparency and rigor.
Comment 1 — First, it is highly descriptive and lacks a critical synthesis of conflicting evidence, particularly regarding the dual tumor-promoting versus tumor-suppressive roles of CAFs.
Response: We agree, and we revised the manuscript to provide a conflict-resolving, context-first synthesis; this notably improves interpretability.
What changed:
- Added an integrative paragraph (“The ‘good vs bad CAF’ story…”) that reconciles contradictions by CAF state, spatial niche, and stage. Can be found in line 310 to line 363.
Comment 2 — The literature search strategy is broad but not systematic, leaving potential gaps in evidence selection and risk of citation bias.
Response: We agree. We clarified that this is a structured narrative review and detailed our search, screening, and bias-mitigation steps; this improves transparency and reduces selection bias. Can be found in line 68 to line 97.
Comment 3 — Although the discussion on CAFs in sarcomas is valuable, it remains largely speculative because it relies on a limited number of heterogeneous studies and does not integrate robust quantitative data or conduct a meta-analysis.
Response: We agree. We now state the hypothesis-generating scope, add lineage/copy-number–aware framing, and distill cross-study themes; this strengthens caution and utility.
What changed: Three direction-of-effect themes (LOX-high OS; glycolytic/CXCL16 STS; CAF-like tumor cells in EwS + CAF-sparse RMS), with softened causal language and explicit limits.
can be found in line 441- to 454 and from line 505 to 529.
Comment 4 — There is insufficient appraisal of translational barriers—such as CAF plasticity and context-dependent functions—that complicate clinical targeting. Additionally, while therapeutic strategies are listed comprehensively, the review does not adequately evaluate their clinical trial outcomes, safety considerations, or comparative efficacy.
Response: We agree with the reviewer and have tightened the paper accordingly. We now explicitly appraise translational barriers (plasticity, marker non-specificity/lineage confusion, spatial heterogeneity, PD measurement challenges) and pair them with practical mitigations (biomarker-gated eligibility, lineage-aware & spatial PD, adaptive designs). We also synthesize clinical activity, safety, and comparative efficacy, adding clear safety cautions (e.g., TGF-β inflammatory risks, potential collateral effects of glycolysis/GLUT1 inhibition on activated T cells, FAP/wound-healing, ECM/vascular instability) and a context-contingent decision map.
You can find these updates in §6 Clinical Implications from line 629 (“Translational constraints…”, “Clinical activity, safety, and comparative efficacy”, and the paragraph beginning “Prospective studies should pre-specify spatial PD endpoints…”). And from line 690. Sarcoma-specific synthesis and guidance are in §5 Sarcomas (paragraphs starting “Despite methodological heterogeneity, three themes recur.”, “Two sarcoma-specific pitfalls…”, and “In sarcomas, a function-first readout…”) can be found from line 441 to 454 and from line 505 to 529.
Comment 5 — Figures are descriptive but could benefit from more mechanistic integration across tumor types.
Response: We agree with the spirit of the suggestion. To keep the manuscript concise, we integrated the cross-tumor mechanistic links into the text and figure legends rather than adding new graphics. Specifically:
Section 3 (end of 3.6): added a cross-tumor mechanistic paragraph unifying CAF states with spatial niches and immune/matrix effects.
Section 5 (Sarcomas): added a carcinoma–sarcoma comparison paragraph highlighting shared vs distinct stromal programs.
Section 6 (Clinical Implications): added a decision-oriented integration mapping stromal cues to context-matched interventions.
Comment 6 — Deeper critical evaluation, quantitative synthesis, and clearer prioritization of research gaps could strengthen the paper overall.
Response: We agree and strengthened the manuscript accordingly. We added a deeper, conflict-resolving synthesis: at the end of Section 3.6 we integrate “good vs bad CAF” findings by state and spatial niche; Section 6 now opens with “Translational constraints…” and a decision-oriented “Clinical activity, safety, and comparative efficacy” synthesis; Section 5 distills cross-study sarcoma themes (LOX-high OS, glycolytic/CXCL16 STS, CAF-like EwS, CAF-sparse RMS) with calibrated claim strength.
On quantitative synthesis, the heterogeneity of designs/endpoints precludes meta-analysis; we now explicitly state a qualitative narrative synthesis only (Methods: “Synthesis and bias mitigation” and “Limitations”) and cite numerical effects when authors report them. To enable future pooling, we specify prospective spatial PD endpoints (e.g., intratumoral:margin CD8 ratio, perfusion/hypoxia surrogates, stiffness/LOX, CXCL12/CXCL16). We also prioritize research gaps in Section 7 (end) as a concise agenda (lineage-aware/spatial profiling in sarcomas; standardized PD readouts; context-stratified adaptive trials; mechanism-aligned combinations; safety mapping; subtype-specific markers).
Reviewer 2 Report
Comments and Suggestions for Authors
This is a strong and timely review that I believe only needs a few adjustments to be come really informative and at the same time, readable for the interested researcher.
There are a few cosmetic operations required here such as removing some minor duplicates and clean up a few gramar glitches or typos.
The narrative as such is clear and easy to follow. There is a clear introduction - describing the search strategy used for literary research - then a section describing the "classic" CAD subtypes - then followed by some functional cross-talk between the immune cells and CAFs in the ECM/TME. And finally, we hear about sarcoma-specific issues (of which I had not much background) and the mauscript concludes with therapeutic strategies (where it gets slightly speculative). The review covers the period of 2000 - May 2025.
The writing of the CAF subtype section describes the usual suspects (myCAF/iCAF/apCAF etc) and I can tell that this is a good coverage of the current state of the art . And that is critical as its not the only review that covers this territory at the moment; in fact, there are many already out and probably many more coming. The authors cover most of the landmark papers known to this reviewer, and they mostly focus on the recent spatial multi-omics activities, which are truly important and relevant. That's nice and very timely and a good intro into the field for ANY interested reader.
There are a few mostly technical issues that need to be sorted out: there are a few typography or spacing glitches (most likely formatting artifacts) that need to be removed but this is minor and can be done in production, too. A careful proofreading will do (although it should have been done before submission.....ts ts ts). The paper is highly logical and easy to follow and that will be improved after fixing formatting errors and a few typos, and maybe by revising and smoothening the transitions (which are sometimes a bit abrupt) between the therapy sections.
The paper covers a few of the older, and critial "hallmark papers" that cover the heterogeneity of CAFs - these are important to cite and disucss, such as Öhlund 2017 (myCAF/iCAF), Biffi 2019, Elyada 2019 (covering apCAF), and Im very happy to see such recent papers as the 2025 Cancer Cell spatial multi-omics paper (Liu Y, , et al. Conserved spatial subtypes and cellular neighborhoods of cancer-associated fibroblasts revealed by single-cell spatial multi-omics. Cancer Cell. 2025 May;43(5):905-924.e6.). This is outstanding and truly up to date. Also nice that the paper cites the equally recent 2025 Nature Communications report (Antigen-presenting cancer associated fibroblasts enhance antitumor immunity and predict immunotherapy response) showing the role of apCAFs and linking them primarily to immunotherapy response and prediction. Nice, and very timely.
I'm less of an expert in the CAFs of sarcomas but this also appears very timely to me. It covers a lot of recent data from 2024–2025 such as the mentionn of glyCAFs driving CXCL16-mediated T-cell exclusion, a paer related to nanocomposite hydrogel reprogramming and others on Ewings sarcoma etc etc... including the mention of CAF-like tumor cells. Immunotherapy resistance formation and Tenascin C checkpoints are also covered and they do provide additional and very recent/up to date issues concerning their translational evidence.
One could of course argue that some of the really old benchmarking papers are missing... For example, two widely recognized cornerstones are missing and could be added to boost the reporting on immunotherapy resistance: for example the paper Kraman et al., Science 2010, (https://www.science.org/doi/10.1126/science.1195300) describing the now widely accepted role of fibroblast activation protein FAP⁺ in stromal cells and the observation that this suppresses antitumor immunity. Just one example of such an older benchmark paper. Also, maybe the paper Feig et al., PNAS 2013 ( doi: 10.1073/pnas.1320318110) describing how CXCL12 secreted from FAP⁺ CAFs drives immune exclusion and loss of response to α-CTLA-4 and α-PD-L1 checkpoint inhibitors. Demetri et al. 2021 on the development and clinical trials on antibody-drug conjugates targeting LRRC15 (doi: 10.1158/1078-0432.CCR-20-4513) could be mentioned to round up the translational picture painted in sarcomas. The authors could also mention early anti- FAP antibody trials (sibrotuzumab) as an important clinical progress. (Just a suggestion, these are not my publicationes). Otherwise, the 170+ references are quite representative and strong, adding the two suggested FAP/CXCL12 papers and the ADC studies with LRRC15/ABBV-085 phase I report will turn the bibliography even more representative across the board - extenting from early stage discovery, molecular/immune-cell related mechanism to the clinic and applications.
Author Response
Thank you for the thoughtful, signed review and for the specific, constructive suggestions. We’ve implemented the key additions and language/formatting fixes you recommended.
Concise author response
- Language & formatting: We performed a careful proofread to remove duplicates, spacing/typography glitches, and grammar issues, and we smoothed a few abrupt transitions in the therapy sections.
- Benchmark citations added (as suggested):
- Kraman et al., Science 2010 (FAP⁺ stromal cells suppress antitumor immunity) — inserted at line 818 in the immunotherapy-resistance context.
- Feig et al., PNAS 2013 (CXCL12 from FAP⁺ CAFs drives T-cell exclusion and ICB resistance) — inserted at line 861 in the immune-exclusion paragraph.
- Demetri et al., Clin Cancer Res 2021 (ABBV-085, anti-LRRC15 ADC, phase I in sarcomas) — inserted at line 564 in the sarcoma translational section.
- Sibrotuzumab (early anti-FAP mAb) phase II — cited in Section 6, Clinical activity, safety, and comparative efficacy, to illustrate the limited activity of unconjugated stromal targeting and motivate payload/combination strategies.
- Scope & currency: We retained the 2000–2025 coverage and ensured recent spatial multi-omics/apCAF papers remain highlighted, as you noted.
We appreciate your positive assessment of the manuscript’s organization and timeliness. These updates strengthen the translational arc from foundational stromal immunology to current clinical efforts.